# MutS/MutL crystal structure reveals that the MutS sliding clamp loads MutL onto DNA

**Flora S Groothuizen[1], Ines Winkler[2], Michele Cristóvão[2], Alexander Fish[1], Herrie HK Winterwerp[1], Annet Reumer[1], Andreas D Marx[2], Nicolaas Hermans[3], Robert A Nicholls[4], Garib N Murshudov[4], Joyce HG Lebbink[3,5], Peter Friedhoff[2], Titia K Sixma[1]\***

[1]Division of Biochemistry and CGC.nl, Netherlands Cancer Institute, Amsterdam, Netherlands; [2]Institute for Biochemistry, Justus-Liebig-University, Giessen, Germany; [3]Department of Genetics, Cancer Genomics Netherlands, Erasmus Medical Center, Rotterdam, Netherlands; [4]Structural Studies Division, MRC Laboratory of Molecular Biology, Cambridge, United Kingdom; [5]Department of Radiation Oncology, Erasmus Medical Center, Rotterdam, Netherlands

**Abstract** To avoid mutations in the genome, DNA replication is generally followed by DNA mismatch repair (MMR). MMR starts when a MutS homolog recognizes a mismatch and undergoes an ATP-dependent transformation to an elusive sliding clamp state. How this transient state promotes MutL homolog recruitment and activation of repair is unclear. Here we present a crystal structure of the MutS/MutL complex using a site-specifically crosslinked complex and examine how large conformational changes lead to activation of MutL. The structure captures MutS in the sliding clamp conformation, where tilting of the MutS subunits across each other pushes DNA into a new channel, and reorientation of the connector domain creates an interface for MutL with both MutS subunits. Our work explains how the sliding clamp promotes loading of MutL onto DNA, to activate downstream effectors. We thus elucidate a crucial mechanism that ensures that MMR is initiated only after detection of a DNA mismatch.

**\*For correspondence:**
t.sixma@nki.nl

**Competing interests:** The authors declare that no competing interests exist.

## Introduction

To enable the correct and complete transfer of genetic information during cell division, DNA polymerases efficiently replicate the genome by pairing nucleotide bases opposite their complementary template base. However, despite the polymerase proofreading ability, incorrect nucleotides are occasionally incorporated into the new DNA strand, resulting in mutations when left uncorrected. To reduce the number of such mismatches and maintain genomic stability, replication is followed by DNA mismatch repair (MMR) in almost all cellular organisms (*Kunkel and Erie, 2005*; *Jiricny, 2013*). The initiation of this MMR system is evolutionarily conserved, although in eukaryotes heterodimeric homologs replace the bacterial homodimeric components. Defects in MMR result in a mutator phenotype and in humans in predisposition for cancer, known as Lynch syndrome or HNPCC (*Lynch and de la Chapelle, 1999*).

MMR is initiated when a MutS homolog binds to a mismatch. In this mismatch recognition step, the MutS dimer kinks the DNA at the site of the mismatch and stacks a phenylalanine onto the mispaired base (*Lamers et al., 2000*; *Obmolova et al., 2000*; *Warren et al., 2007*). Upon ATP binding MutS releases the mismatch (*Allen et al., 1997*; *Gradia et al., 1997*) and travels as a 'sliding clamp' along the DNA helix (*Gradia et al., 1999*; *Acharya et al, 2003*; *Jeong et al., 2011*), and this specific state of

**eLife digest** The genetic code of DNA is written using four letters: "A", "C", "T", and "G". Molecules of DNA form a double helix in which the letters in the two opposing strands pair up in a specific manner—"A" pairs with "T", and "C" pairs with "G". A cell must replicate its DNA before it divides, and sometimes the wrong DNA letter can get added into the new DNA strand. If left uncorrected, these mistakes accumulate over time and can eventually harm the cell. As a result, cells have evolved several ways to identify these mistakes and correct them, including one known as "mismatch repair".

Mismatch repair occurs via several stages. The process starts when a protein called MutS comes across a site in the DNA where the letters are mismatched (for example, where an "A" is paired with a "C", instead of a "T"). MutS can recognize such a mismatch, bind it, and then bind to another molecule called ATP. MutS then changes shape and encircles the DNA like a clamp that can slide along the DNA. Only when it forms this "sliding clamp" state can MutS recruit another protein called MutL. This activity in turn triggers a series of further events that ultimately correct the mismatch. However, it remains poorly understood how MutS forms a clamp around DNA and how and why this state recruits MutL in order to start the repair.

To visualize this short-lived intermediate, Groothuizen et al. trapped the relevant complex in the presence of DNA containing a mismatch and then used a technique called X-ray crystallography to determine the three-dimensional structure of MutS bound to MutL. The structure reveals that two copies of MutS tilt across each other and open up a channel, which is large enough to accommodate the DNA. In this manner, MutS is able to form a loose ring around the DNA. The changes in the structure and the movement of the DNA to the new channel were confirmed using another technique, commonly referred to as FRET.

Groothuizen et al. observed that the movements in the MutS protein also serve to make the interfaces available that can recognize MutL. If these interfaces were disturbed, MutS and MutL were unable to associate with each other, which resulted in a failure to trigger mismatch repair. Further analysis revealed that that MutL binds to DNA only after MutS has recognised the mismatch and formed a clamp around it. This is the first time that the MutS clamp and the MutS/MutL complex have been visualized, and further work is now needed to understand how MutL triggers other events that ultimately repair the mismatched DNA.

MutS is recognized by MutL or its homologs (*Grilley et al., 1989*; *Prolla et al., 1994*; *Drotschmann et al., 1998*; *Acharya et al., 2003*).

MutL proteins are constitutive dimers through their C-terminal domains, while the N-terminal ATPase domains reorganize and dimerize upon ATP binding (*Grilley et al., 1989*; *Ban and Yang, 1998*; *Ban et al., 1999*; *Guarné et al., 2004*). Once recruited by the MutS sliding clamp, the MutL homologs activate downstream repair. This includes the nicking of the newly replicated strand by a nuclease, which is either part of the MutL C-terminal domain (*Kadyrov et al., 2006*), or a separate protein such as MutH in *Escherichia coli* (*Hall and Matson, 1999*). MutL also activates UvrD in bacteria to unwind the DNA (*Yamaguchi et al., 1998*), after which the new DNA strand can be removed and re-replicated (*Kunkel and Erie, 2005*).

As loss of MutS homologs (MSH2, MSH3 and MSH6 in humans) or MutL homologs (MLH1 and PMS2 in humans) leads to mutator and/or cancer phenotypes, these proteins evidently have critical roles in mismatch repair and it is therefore important to understand their exact mechanism. Despite extensive studies (*Gradia et al., 1999*; *Mendillo et al., 2005*; *Cho et al., 2012*; *Qiu et al., 2012*), it is unclear how MutS achieves the sliding-clamp state, how this promotes MutL recognition and why this results in activation of the MutL protein.

Here, we trap the transient complex between MutS and MutL to resolve a crystal structure of the MutS sliding clamp bound to MutL. This is, to our knowledge, the first time that not only this MutS conformation but also the complex between MutS and MutL could be observed. We show how rearrangements in MutS promote interactions from both MutS subunits with a single MutL N-terminal domain, and how this domain is then positioned to load onto DNA running through a novel channel in the MutS dimer. We use biophysical methods to analyze the transient states and mechanistically

understand the specificity and effect of MutL binding to MutS, and functional assays to address how this affects MMR initiation.

## Results

### Structure of the MutS/MutL complex

To trap the *E. coli* MutS/MutL complex we used site-specific chemical crosslinking of single-cysteine variants of MutS and MutL, with a flexible BM(PEO)$_3$ crosslinker. First all cysteines in MutS and MutL were replaced and functionality of the resulting protein was confirmed (*Giron-Monzon et al., 2004*; *Manelyte et al., 2006*; *Winkler et al., 2011*). Then single cysteines were introduced to find positions where crosslinking was dependent on sliding clamp formation. MutS D246C crosslinks specifically to MutL N131C only when a DNA mismatch and a nucleotide are present (*Winkler et al., 2011*; *Figure 1A*, *Figure 1—figure supplement 1A*), indicating that a complex relevant for MMR is trapped.

For structural studies, we scaled up the reaction and removed C-terminal domains from MutS and MutL (*Figure 1A*), to capture the complex between MutS$^{\Delta C800}$ D246C (which we will refer to as MutS$^{\Delta C800}$) and the 40 kDa N-terminal LN40 domain (*Ban and Yang, 1998*) of MutL N131C (which we will refer to as MutL$^{LN40}$). The proteins were crosslinked in the presence of mismatched DNA and ATP, followed by purification to obtain the protein, and then this cycle was repeated in order to obtain fully crosslinked material. This generated a complex where each MutS$^{\Delta C800}$ subunit in the dimer binds to a MutL$^{LN40}$ monomer (*Figure 1A*, *Figure 1—figure supplement 1B,C*), which was sufficiently homogeneous and stable to allow crystallization.

We crystallized the MutS$^{\Delta C800}$/MutL$^{LN40}$ complex in the presence of DNA containing a G:T mismatch and the non-hydrolyzable ATP analog AMP-PNP (adenylyl-imidodiphosphate). The complex crystallized in several different space groups, diffracting to resolutions from 7.6 to 4.7 Å. In all crystal forms, we could elucidate the same structure of the protein complex (*Figure 1B*, *Figure 1—figure supplements 2*, *Table 1*), using parts of higher-resolution MutS$^{\Delta C800}$ and MutL$^{LN40}$ structures for molecular replacement.

The crystal structure shows a novel conformation of MutS, in which the subunits in the dimer are tilted across each other by ∼30°, compared to the mismatch recognition state (*Figure 1C,D*). The subunits are tilted as a rigid body, but the C-terminal HTH domains hinging around residues 765–766, move with the opposite subunit, maintaining their role in stabilizing MutS dimers (*Biswas et al., 2001*). Meanwhile, the connector domains are rotated by ∼160° around a hinge at residues 265–266, which moves these domains out of the center of the molecule and packs them against the ATPase domains (*Figure 1C,E*). The mismatch-binding domain could not be resolved in the density, probably because it is flexible in this state. While the mismatch recognition state of MutS is asymmetric (*Lamers et al., 2000*), this MutL$^{LN40}$-bound conformation shows a more symmetrical MutS$^{\Delta C800}$ dimer.

The MutL$^{LN40}$ interaction with MutS$^{\Delta C800}$ involves two interfaces (*Figure 1F*). The first interface is formed by the largest β-sheet of the ATPase domain of MutL$^{LN40}$, and the ATPase and core domains of one subunit of MutS$^{\Delta C800}$. The second interface involves the side of this same β-sheet and a looped-out helix of MutL$^{LN40}$, and the newly positioned connector domain of the other MutS$^{\Delta C800}$ subunit. Each MutL$^{LN40}$ monomer is therefore interacting with both subunits in the MutS$^{\Delta C800}$ dimer.

### Conformation of the MutS sliding clamp

The novel conformation of MutS in our crystal structure reveals a rearrangement of the subunits in the MutS$^{\Delta C800}$ dimer, tilting around the interface formed by the two ATPase sites (*Figure 1D*, *Video 1*). The tilting creates a new MutS dimer interface of ∼500 Å$^2$ where the clamp domains cross over, partially from interactions between the helices themselves (200 Å$^2$), the rest from the ends of the clamp domains with the helices.

We observe nucleotide density in the ATP binding sites of both subunits in the MutS$^{\Delta C800}$ dimer (*Figure 2—figure supplement 1A*), and since we crystallized the protein with AMP-PNP we modeled these nucleotides in the density. This type of ATP-induced tilting and increased packing of ATPase domains is more often observed upon ATP binding in ABC ATPases, such as ATP transporters, SMCs and RAD50 (*Hopfner and Tainer, 2003*). Based on comparison to RAD50 (*Hopfner et al., 2000*) we previously predicted a tilting motion (*Lamers et al., 2004*), and an open-to-closed transition has been supported by deuterium exchange mass spectrometry (*Mendillo et al., 2010*), but the crossing of the clamp domains of MutS and the effect that this has on DNA binding were unexpected.

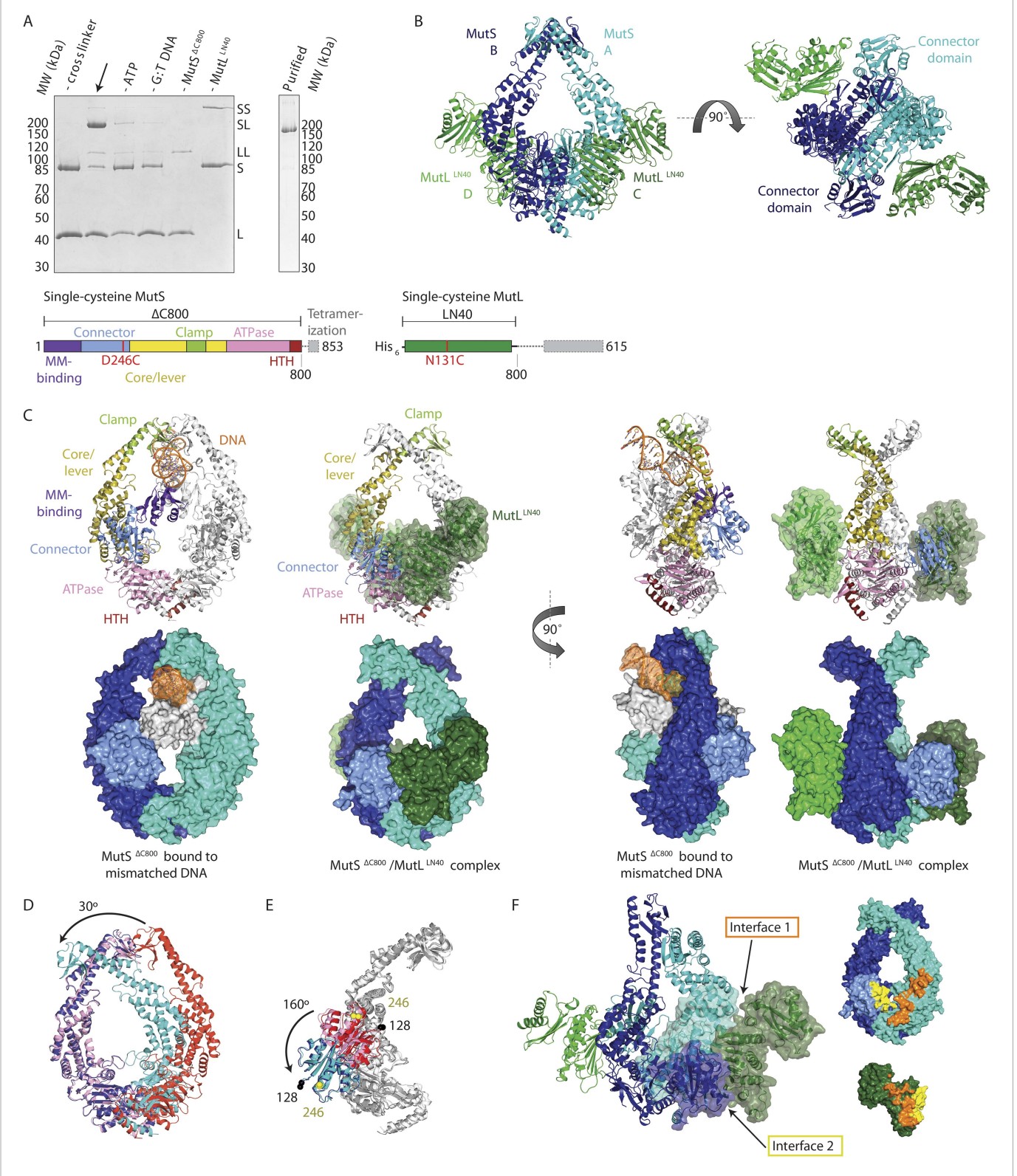

**Figure 1**. Crystal structure of the crosslinked MutS^ΔC800/MutL^LN40 complex. (**A**) DNA and ATP-dependent crosslinking of MutS^ΔC800 D246C (S) and MutL^LN40 N131C (L) and large-scale purification. Constructs and domain definitions are shown. (**B**) Crystal structure of the trapped transient complex of MutS^ΔC800 dimer (blue/cyan) with MutL^LN40 (green). (**C**) Comparison between MutS^ΔC800 in mismatch-recognition state (1E3M.pdb) and the MutS^ΔC800/MutL^LN40 complex, with MutS subunit B colored as in (**A**). (**D**) The dimer subunits (blue/cyan) tilt across each other (connector and mismatch-binding domains not

Figure 1. Continued

shown for clarity) compared to the mismatch-bound state (red/pink). (**E**) The connector domain (blue/cyan) rotates around residues 265–266 compared to the mismatch-bound state (red/pink) relative to other domains. Reorientation of residues 128 and 246 indicated. (**F**) Each MutL$^{LN40}$ subunit (green) interacts via two interfaces (orange/yellow) with the MutS$^{\Delta C800}$ dimer (blue/cyan).

The following figure supplements are available for figure 1:

**Figure supplement 1**. Crosslinking, purification and crystal structure of the 856 MutS$^{\Delta C800}$/MutL$^{LN40}$ complex.

**Figure supplement 2**. Electron density for different crystal forms of the MutS$^{\Delta C800}$/MutL$^{LN40}$ complex.

The type of rearrangement of the MutS N-terminal region was similarly unexpected. In this movement the connector domains have rotated onto the so-called 'signature helix' (residues 670–684) (*Hopfner and Tainer, 2003*), whose amino terminus interacts with the γ-phosphate of the ATP in the opposite subunit in ABC ATPases. Therefore the observed connector domain movement could be the result of binding of ATP in the opposing subunit.

In RAD50 this tilting or 'closing' motion is transmitted through a 'signature coupling helix' via charged interactions with the signature helix (*Williams et al., 2011*; *Deshpande et al., 2014*). This 'coupling helix' is found at the beginning of a long stretch (144–767) in RAD50 that includes the coiled coil region and ends in the signature helix. The equivalent region in MutS is only 10 residues long (660–669) and it is disordered in all structures. It is feasible that this 10-residue loop is critical for transmission of the ATP signal, but the details must be different, since the basic residues in the signature helix of RAD50 are not conserved in MutS.

To validate the relevance of the observed conformational changes for the MMR process, MutS proteins with a single cysteine at position 449 were site-specifically labeled with two different Alexa fluorophores and combined into heterodimers by random subunit exchange (*Figure 2A*, *Figure 2—figure supplement 2*). When labeled protein was bound to end-blocked DNA containing a G:T mismatch, FRET increased upon ATP addition. This indicates that ATP-induced sliding clamp formation moves these residues toward each other, in line with the shorter distance in the new conformation (from 50 Å in the mismatch-recognition state to 43 Å in the MutL$^{LN40}$-bound structure).

The new position of the connector domain brings it closer to the ATPase domain (*Figure 1E*, *Video 1*). To analyze this movement we combined two single-cysteine variants of MutS, labeled in the connector domain (residue 246) and the ATPase domain (residue 798) respectively, into heterodimers, and measured the FRET signal between these sites upon sliding clamp formation (*Figure 2B*, *Figure 2—figure supplement 2*). Indeed, after ATP addition the FRET increased, indicating that these residues come closer together. As this is measured in the absence of MutL it suggests that after mismatch binding, ATP is sufficient to induce movement of the connector domain away from the mismatch-recognition position.

Although the complex was crystallized in the presence of DNA containing a mismatch, the DNA is not visible in the structure. This could be due to smearing out of the electron density over multiple positions or the DNA may not be present in the crystal, both indicating that the mismatch has been released, as expected for the ATP-bound state of MutS.

The subunit tilting has occluded the original DNA binding site, but because the connector and mismatch-binding domains have moved, a large channel (∼35 Å wide) in MutS has become accessible, which could easily accommodate a DNA duplex (20 Å diameter). The new channel is lined by conserved lysines and arginines (*Figure 2C*, *Figure 2—figure supplement 1C*), which can govern nonspecific contacts with the negative backbone of DNA, as expected for the MutS sliding clamp state (*Cho et al., 2012*). Moreover, in our crystal forms these channels are aligned between symmetry mates or even within the asymmetric unit (*Figure 2—figure supplement 1B*). This packing of MutS/MutL complexes is most likely a crystallographic artefact, as it could not occur in the presence of MutL dimers, but the alignment could reflect the path of the DNA present during crystallization. We hypothesize that the DNA is pushed down to this channel during the ATP-induced conformational changes of MutS after mismatch recognition.

To test whether DNA moves down into the new channel in solution, we analyzed FRET signals between fluorescently labeled DNA (end-blocked) and specific sites in single-cysteine MutS variants

**Table 1**. Data collection and refinement statistics

| | Crystal form 1 27-bp DNA | Crystal form 2 27-bp DNA | Crystal form 3 100-bp DNA |
|---|---|---|---|
| Data collection | | | |
| Space group | C2 | C2 | P2$_1$ |
| Cell dimensions | | | |
| $a$, $b$, $c$ (Å) | 165.9, 188.5, 200.4 | 380.6, 126.5, 243.3 | 192.6, 109.4, 277.5 |
| $\alpha$, $\beta$, $\gamma$ (°) | 90.0, 94.8, 90.0 | 90.0, 91.4, 90.0 | 90.0, 90.0, 90.0 |
| Resolution (Å)* | 82.7–4.71 (4.96–4.71) | 49.94–6.6 (7.13–6.6) | 49.3–7.6 (8.5–7.6) |
| $R_{merge}$ | 19.4 (79.7) | 21.3 (80.1) | 16.8 (91.9) |
| $I/\sigma I$ | 2.5 (1.0) | 3.4 (1.1) | 4.3 (1.0) |
| Completeness (%) | 97.3 (98.0) | 96.8 (97.7) | 81.3 (82.5) |
| Redundancy | 2.4 (2.4) | 2.9 (3.0) | 2.3 (2.2) |
| Refinement | | | |
| Resolution (Å) | 4.7 | 6.6 | 7.6 |
| No. reflections | 31,052 | 21,305 | 11,763 |
| $R_{work}$/ $R_{free}$ | 31.8/35.0 | 25.6/28.7 | 26.2/30.5 |
| No. atoms | 21,906 | 45,054 | 45,054 |
| Protein | 21,813 | 44,868 | 44,868 |
| Ligand/ion | 93 | 186 | 186 |
| Water | 0 | 0 | 0 |
| B-factors | | | |
| Protein | 212 | 255 | 221 |
| Ligand/ion | 220 | 212 | 171 |
| Water | n/a | n/a | n/a |
| R.m.s deviations | | | |
| Bond lengths (Å) | 0.009 | 0.0103 | 0.0113 |
| r.m.s. Z (bonds) | 0.45 | 0.51 | 0.55 |
| Bond angles (°) | 1.32 | 1.35 | 1.31 |
| r.m.s. Z (angles) | 0.59 | 0.70 | 0.68 |

*Highest resolution shell is shown in parenthesis.

(*Figure 2D*). After addition of ATP, DNA moves away from residues 449 at the DNA-clamp position (FRET/acceptor ratio reduction ~1.5 fold), while an increase in FRET/acceptor ratio (>3.6 fold) was observed when MutS was labeled at position 336. Since the connector domain moves down itself, no substantial change in FRET/acceptor ratio is observed between residue 246 and DNA (*Figure 2—figure supplement 1D*, *Figure 2—figure supplement 2*). Combined, these FRET data are in agreement with repositioning of the DNA towards the channel created by the new conformation.

Based on these validations, we conclude that the observed MutS conformation in our crystal structure is induced by ATP after mismatch recognition. Since the new position of the DNA would allow MutS to fit as a loose ring around the DNA duplex (with a channel size similar to that of PCNA (*Krishna et al., 1994*)), consistent with free movement over DNA (*Cho et al., 2012*), we propose that this is the MutS sliding clamp conformation.

## Orientation of MutL$^{LN40}$ on MutS

In the structure MutL$^{LN40}$ makes two interfaces with MutS$^{\Delta C800}$. Interface 1 orients MutL$^{LN40}$ on the ATPase and core domains of MutS. Recently, a loop in *Bacillus subtilis* MutS was found to be essential for MutL interaction (*Lenhart et al., 2013*). Although the equivalent loop is shorter in *E. coli* MutS and

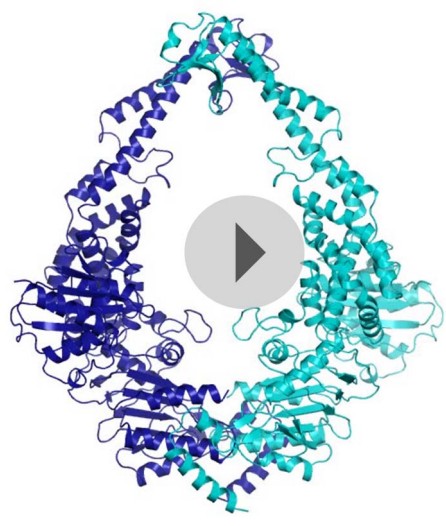

**Video 1.** Interpolation between two MutS conformations. Interpolation between the mismatch-bound conformation of MutS and the conformation as observed in complex with MutL$^{LN40}$ shows tilting of the MutS subunits across each other. The connector domain rotates outward, although the exact trajectory may be different than in this visualization. Mismatch-recognition domains are not shown since they are not visible in the MutS$^{\Delta C800}$/MutL$^{LN40}$ structure.

the explicit residues (F319/F320) are missing, the corresponding region is located within the ~590 Å$^2$ interface (interface 1) with MutL$^{LN40}$.

We validated the observed interaction at interface 1 by a crosslinking experiment with a short crosslinker. We created single-cysteine mutants MutS$^{\Delta C800}$ A336C and MutL$^{LN40}$ T218C (*Figure 3A*), which are located ~7.4 Å apart in the structure, and then showed that we could crosslink them efficiently with a short cysteine-specific crosslinker (8 Å, BMOE), dependent on the presence of both mismatched DNA and ATP (*Winkler et al., 2011*). Only background crosslinking occurred when using MutS$^{\Delta C800}$ D246C (connector domain) with MutL$^{LN40}$ T218C (interface 1) under these conditions (*Figure 3—figure supplement 1A*), indicating that the crosslinking between MutS$^{\Delta C800}$ A336C and MutL$^{LN40}$ T218C is specific.

To further verify interface 1 between MutS and MutL, we tested whether mutations in the interface affected MMR activity in vivo, in a complementation assay with MutS or MutL deficient cells (*Figure 3B,F*, *Table 2*). We found several mutants of MutL (A138E, A138E/H139A, R55D/R57D, or combinations) and a triple mutant in MutS (P595A/I597A/M759D) that could not complement loss of wild type (WT) protein. We purified the mutants that impaired MMR and characterized their defects. The MutS triple mutant has a slight defect in ATPase activity but this does not impair its sliding clamp formation (*Figure 3—figure supplement 1B,C*), and other mutants with similar ATPase effects (e.g. MutS F596A) can almost fully reconstitute MMR (*Junop et al., 2003*), suggesting that the in vivo effect we observe is due to the perturbed interface with MutL.

To assess the effect of these mutations on binding of MutL to the transient MutS sliding clamp we designed a two-stage assay using Surface Plasmon Resonance (SPR). We first formed and trapped MutS sliding clamps on 100-bp end-blocked DNA in the presence of ATP (*Groothuizen et al., 2013*). Next, MutL was injected, which could then bind to these MutS clamps. By subtraction of the MutS signal, the contribution of MutL could be evaluated for the different mutants (*Figure 3C*), since MutL alone shows no DNA binding under these conditions (*Figure 3—figure supplement 1D*). Indeed the interface 1 mutants that were deficient for MMR conferred a deficiency in MutS/MutL complex formation (*Figure 3D*).

## MutS sliding clamp recognition by MutL

The rearrangement of the connector domain creates a second interface with MutL$^{LN40}$ (interface 2, *Figure 3E*). Previous deuterium exchange experiments (*Mendillo et al., 2009*) indicated that the connector domain interacts with MutL, particularly via MutS glutamines 211 and 212. Indeed in our structure these residues are buried within this ~670 Å$^2$ interface with MutL$^{LN40}$ (*Figure 3F*). Interestingly, the deuterium exchange experiments identified a second region on the MutS surface that was protected upon MutL interaction in the ATPase domain (residues 673–686). These residues are located in the 'signature helix' of MutS (*Hopfner and Tainer, 2003*) and in the complex structure this region is masked by the MutS$^{\Delta C800}$ connector domain in its new position (*Figure 3—figure supplement 1E*).

ATP binding is sufficient to displace the connector domain (*Figure 2B*), and MutL$^{LN40}$ interaction may stabilize the position of the connector domain that we see in the crystal structure. At the resolution of our structure, there is no clear electron density for the connecting crosslinker that we

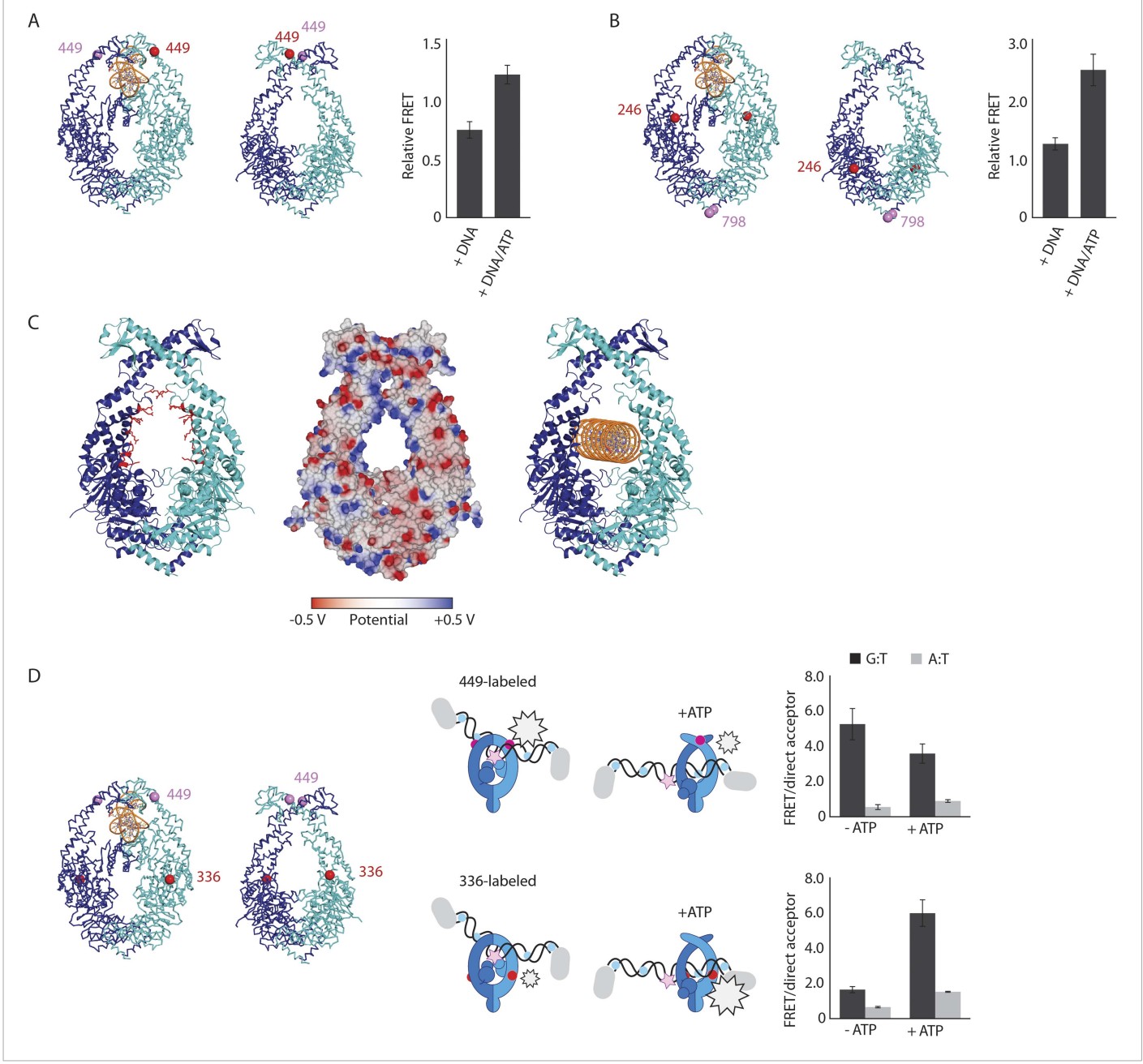

**Figure 2**. The structure of the MutS$^{\Delta C800}$/MutL$^{LN40}$ complex reveals the MutS sliding clamp conformation. (**A**) FRET within MutS dimers (normalized for unbound protein) reveals residues 449 coming closer together upon ATP addition. Error bars depict mean ± SD, n = 3. (**B**) FRET assay agrees with residue 246 on the connector domain of MutS moving towards residue 798 upon ATP addition after mismatch recognition. (**C**) Mismatch and ATP-induced conformational changes open a channel lined by positively charged residues (left: arginines and lysines as red sticks, middle: electrostatic surface), which would fit a DNA helix (right). (**D**) FRET assay agrees with movement of DNA away from residues 449 in MutS, while approaching residues 336 upon ATP addition as schematically depicted (DNA mismatch represented by pink star).

The following figure supplements are available for figure 2:

**Figure supplement 1**. ATP-analog and DNA in the crystal structure.

**Figure supplement 2**. FRET assay – controls and raw data.

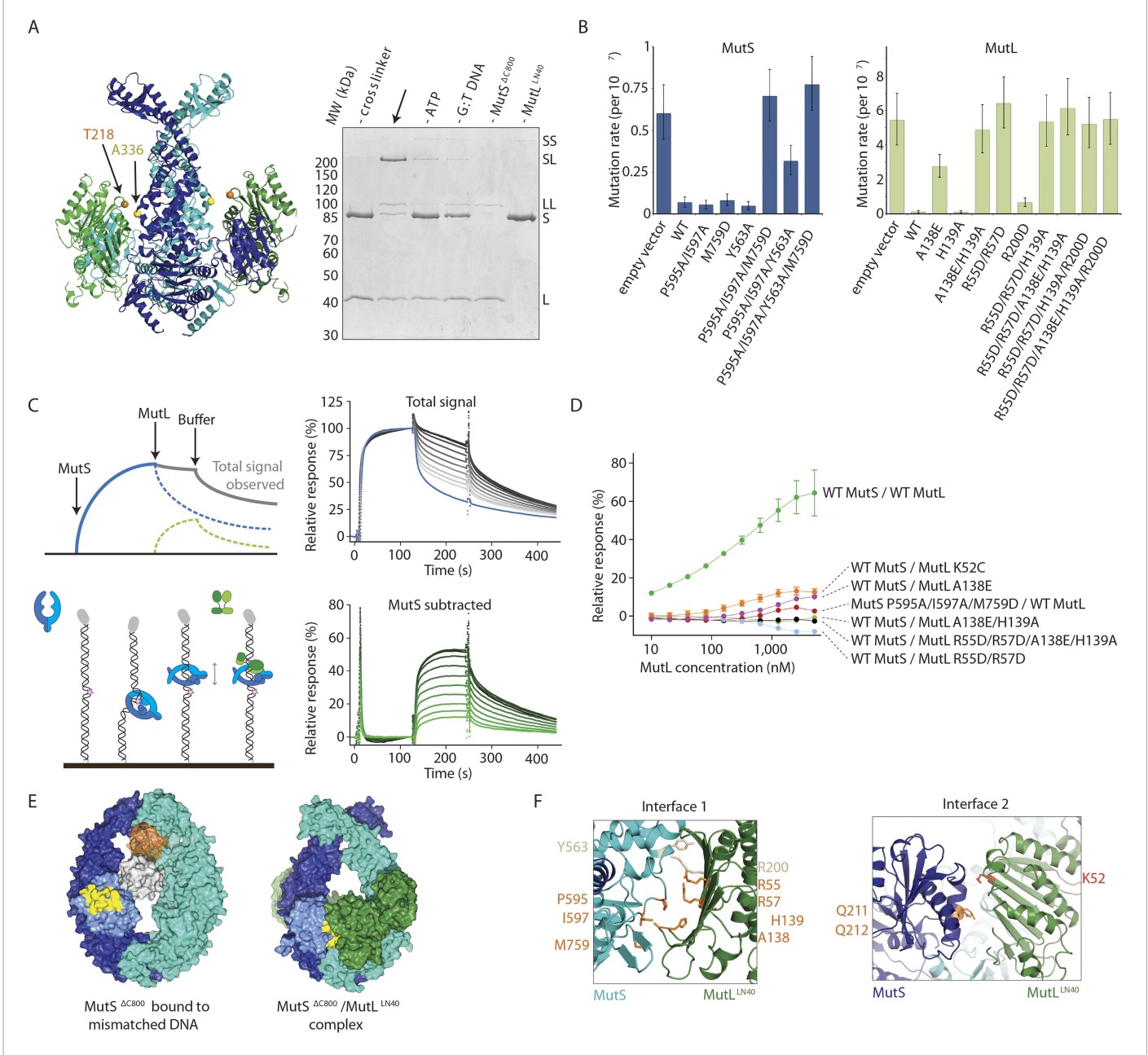

**Figure 3**. Interaction of the MutS$^{\Delta C800}$ sliding clamp with MutL$^{LN40}$. (**A**) Crosslinking occurs between MutS$^{\Delta C800}$ A336C and MutL$^{LN40}$ T218C using BMOE (right panel), as suggested by the structure (left panel). (**B**) Spontaneous mutation rates after complementing MutS or MutL-deficient cells with the indicated mutants. Error bars represent 95% confidence intervals. (**C**) SPR assay to measure MutL binding to pre-formed MutS sliding clamps on end-blocked DNA. MutL contribution (green dotted line) is approached by subtracting MutS-only contribution (blue line) from the total signal (solid line). Data normalized to maximum MutS response. (**D**) MutL and MutS mutants with deficiency in MMR show reduced MutS/MutL complex formation in SPR. Error bars represent SD for averages between two experiments. (**E**) The yellow patch of MutS$^{\Delta C800}$ interacts with MutL$^{LN40}$ in the new conformation after rearrangement of the connector domain. (**F**) Residues in MutS$^{\Delta C800}$/MutL$^{LN40}$ interfaces. Red: full MMR deficiency upon mutation; orange: deficiency upon combination; white: mild effect.

The following figure supplements are available for figure 3:

**Figure supplement 1**. MutS–MutL interaction.

**Figure supplement 2**. (**A**) MutL$^{LN40}$ (L) coelutes with crosslinked MutS$^{\Delta C800}$/MutL$^{LN40}$ complex (SL) from size-exclusion chromatography (right), after incubation with 100-bp DNA with a G:T mismatch and AMP-PNP, indicating that MutL can still dimerize in this complex.

**Table 2.** Mutation rates for MutS and MutL mutants as determined using in vivo complementation assays

| Protein | Mutations per $10^7$ | (95% confidence interval) |
|---|---|---|
| **MutS variant (MutL interface)** | | |
| Empty vector | 0.601 | (0.446–0.772) |
| WT MutS | 0.0686 | (0.0408–0.101) |
| MutS P595A/I597A | 0.0545 | (0.0310–0.0826) |
| MutS M759D | 0.0819 | (0.0490–0.121) |
| MutS Y563A | 0.0488 | (0.0272–0.0749) |
| MutS P595A/I597A/M759D | 0.704 | (0.556–0.864) |
| MutS Y563A/P595A/I597A | 0.317 | (0.233–0.411) |
| MutS Y563A/P595A/I597A/M759D | 0.773 | (0.618–0.941) |
| **MutL variant (MutS interface)** | | |
| Empty vector | 5.43 | (4.00–7.00) |
| WT His-MutL | 0.121 | (0.0542–0.206) |
| His-MutL A138E | 2.76 | (2.12–3.46) |
| His-MutL H139A | 0.103 | (0.0439–0.179) |
| His-MutL A138E/H139A | 4.87 | (3.55–6.33) |
| His-MutL R55D/R57D | 6.41 | (4.99–7.95) |
| His-MutL R200D | 0.663 | (0.432–0.932) |
| His-MutL R55D/R57D/H139A | 5.33 | (3.93–6.89) |
| His-MutL R55D/R57D/A138E/H139A | 6.13 | (4.58–7.84) |
| His-MutL R55D/R57D/H139A/R200D | 5.22 | (3.84–6.76) |
| His-MutL R55D/R57D/A138E/H139A/ R200D | 5.48 | (4.04–7.06) |
| **MutL variant (DNA binding)** | | |
| His-MutL R266E | 5.87 | (4.78–7.04) |
| His-MutL R162E/R266E/R316E | 5.39 | (4.37–6.49) |

Mutation rates and 95% confidence intervals were determined using the Fluctuation AnaLysis CalculatOR (http://www.mitochondria.org/protocols/FALCOR.html) using the MSS-MLE method. For MutS, at least 24 independent colonies were picked; for MutL at least 12 independent colonies were picked.

used to stabilize the complex, and the crosslinked residue 131C on MutL^LN40 could not be modeled. However, the distance between Cα atoms of crosslinked residue 246C in MutS^ΔC800 and residue 132 in MutL^LN40 is shorter (~15.5 Å) than the 18 Å crosslinker (further spaced by cysteine side-chains), showing that the crosslinker can not enforce the observed position of the connector domain.

On the MutL side of interface 2, residue K52 of MutL^LN40 is involved in the interaction with the connector domain of MutS^ΔC800 (**Figure 3F**). This explains the previously reported unexpected mutator phenotype of MutL K52C (**Giron-Monzon et al., 2004**). To confirm its role in the interface we measured the binding of MutL K52C to the MutS sliding clamp in our SPR assay (**Figure 3D**). Indeed, the binding of this mutant is reduced compared to WT MutL.

The ATP-induced tilting of the subunits within MutS and the accompanying connector domain movement positions interfaces 1 and 2 such that they become simultaneously available for binding to the N-terminal domain of MutL (**Figure 1F**). Perturbing either interface 1 or interface 2 impairs MutL binding and MMR (**Figure 3F**). This explains the specificity of MutL for the MutS sliding clamp, which has never been understood before.

MutL proteins dimerize through the C-terminal LC20 domains. The LN40 domains are monomeric in isolation, but can form unstable dimers after ADP or ATP binding or stable dimers when incubated with AMP-PNP (**Ban and Yang, 1998**; **Ban et al., 1999**). Our crosslinked protein crystallizes as MutS^ΔC800 dimers bound to MutL^LN40 monomers, and does not show the MutL^LN40 dimer arrangement

through crystal contacts. Accordingly the MutL monomers have the apo-conformation of residues 80–103 (*Ban and Yang, 1998*) and no density for a nucleotide is visible. However, the interfaces with MutS sterically allow MutL dimerization (*Figure 3—figure supplement 1F*), and in analytical gel filtration, MutL$^{LN40}$ coelutes with the S$_2$/L$_2$ complex after incubation with DNA and AMP-PNP (*Figure 3—figure supplement 2A*).

The stoichiometry of the MutS/MutL complex in vivo is a topic of interest (*Hombauer et al., 2011*; *Elez et al., 2012*). To obtain crystallizable complexes, MutL$^{LN40}$ was bound to each MutS$^{\Delta C800}$ subunit in our experiments, but during MMR a symmetric complex may not be necessary. Indeed the asymmetry of the eukaryotic MMR proteins suggest that this is not required and that a single heterodimeric MutLα will bind to one MSH2/MSH6 or MSH2/MSH3 heterodimer. Literature suggests that interface 2 will be made with MSH2 (*Mendillo et al., 2009*), implying that interface 1 will be with MSH6. The observed MutL$^{LN40}$ protein would then correlate with the MLH1 subunit (*Plotz et al., 2003*) (*Figure 3—figure supplement 2B*).

## Binding to MutS positions MutL on DNA

MutL and homologs have weak DNA binding ability (*Bende and Grafström, 1991*; *Ban et al., 1999*; *Hall et al., 2001*; *Plotz et al., 2003*) which is only clearly observed in low salt conditions, and retention of MutS on DNA upon MutL interaction has been observed (*Drotschmann et al., 1998*; *Schofield et al., 2001*). Although different from the proposed DNA orientation in the crystal (*Figure 2—figure supplement 1B*), a model can be constructed in which the DNA running through the channel in the MutS sliding clamp is simultaneously bound by the proposed DNA binding grooves of the MutL$^{LN40}$ subunits (*Schorzman et al., 2011*) (*Figure 4A*, *Figure 4—figure supplement 1A*). While such DNA binding may require additional conformational changes of MutL, it suggests a mechanism where MutS loads MutL onto DNA.

We tested for MutL$^{LN40}$ loading onto DNA in the context of the MutS/MutL complex in an SPR assay, comparing MutS$^{\Delta C800}$ alone with MutS$^{\Delta C800}$ crosslinked to MutL$^{LN40}$ when it is flowed over 100-bp DNA with a G:T mismatch in the presence of ATP (*Figure 4B*). MutS$^{\Delta C800}$ alone displays fast release from the DNA due to ATP-dependent sliding-clamp formation (*Groothuizen et al., 2013*), as shown by the effect of blocking the end of the DNA (*Figure 4—figure supplement 1E*). The presence of crosslinked MutL$^{LN40}$ greatly reduces the rate of release, suggesting additional DNA binding. The magnitude of the signal in response units on a 41-bp oligomer shows that a single MutS$^{\Delta C800}$/MutL$^{LN40}$ complex is sufficient for this effect (*Figure 4—figure supplement 1B,C*). This delay in release from DNA is also observed when using a mixture of WT MutL and MutS$^{\Delta C800}$, although to a lesser extent (*Figure 4—figure supplement 1D*). The remaining slow release of the crosslinked complex is not affected by blocking of the free DNA end by antibody (*Figure 4—figure supplement 1E*) indicating that the constitutive interaction with crosslinked MutL$^{LN40}$ completely stops MutS$^{\Delta C800}$ dissociation from DNA ends.

To validate that the slower release from DNA is indeed due to MutL$^{LN40}$ binding to DNA, we made point mutants of the MutL$^{LN40}$ protein and crosslinked them to MutS. Mutation R266E reduces DNA binding by MutL (*Junop et al., 2003*; *Robertson et al., 2006*) (*Figure 4B*), most pronounced in full-length context. This mutation also reduces the ability of crosslinked MutL$^{LN40}$ to retain the MutS$^{\Delta C800}$ sliding clamp on DNA (*Figure 4D*, *Figure 4—figure supplement 1B,F*). When introducing two additional mutations (R162E and R316E) in the MutL$^{LN40}$ DNA binding site as suggested by the crystal structure (*Figure 4A*), DNA binding is completely abolished (*Figure 4C*) and the MutS$^{\Delta C800}$/MutL$^{LN40}$ complex releases as fast as MutS$^{\Delta C800}$ alone (*Figure 4D*, *Figure 4—figure supplement 1B,F*). This indicates that MutL binds DNA when interacting with the MutS sliding clamp.

## MutL is loaded onto DNA after MutS releases the mismatch, which is essential in MMR

To assess whether the loading of MutL$^{LN40}$ onto DNA is kinetically distinct from MutS mismatch recognition, we set up an assay to separate events. We read out mismatch recognition (*Lamers et al., 2000*; *Obmolova et al., 2000*; *Warren et al., 2007*) by the kinking of DNA, which can be assessed using 45-bp heteroduplex DNA labeled with Alexa fluorophores on each side of the mismatch (*Cristóvão et al., 2012*), *Figure 5A*, *Figure 5—figure supplement 1A*), in a stopped-flow set up. In parallel we follow DNA interaction using fluorescence polarization (FP) of TAMRA-labeled DNA

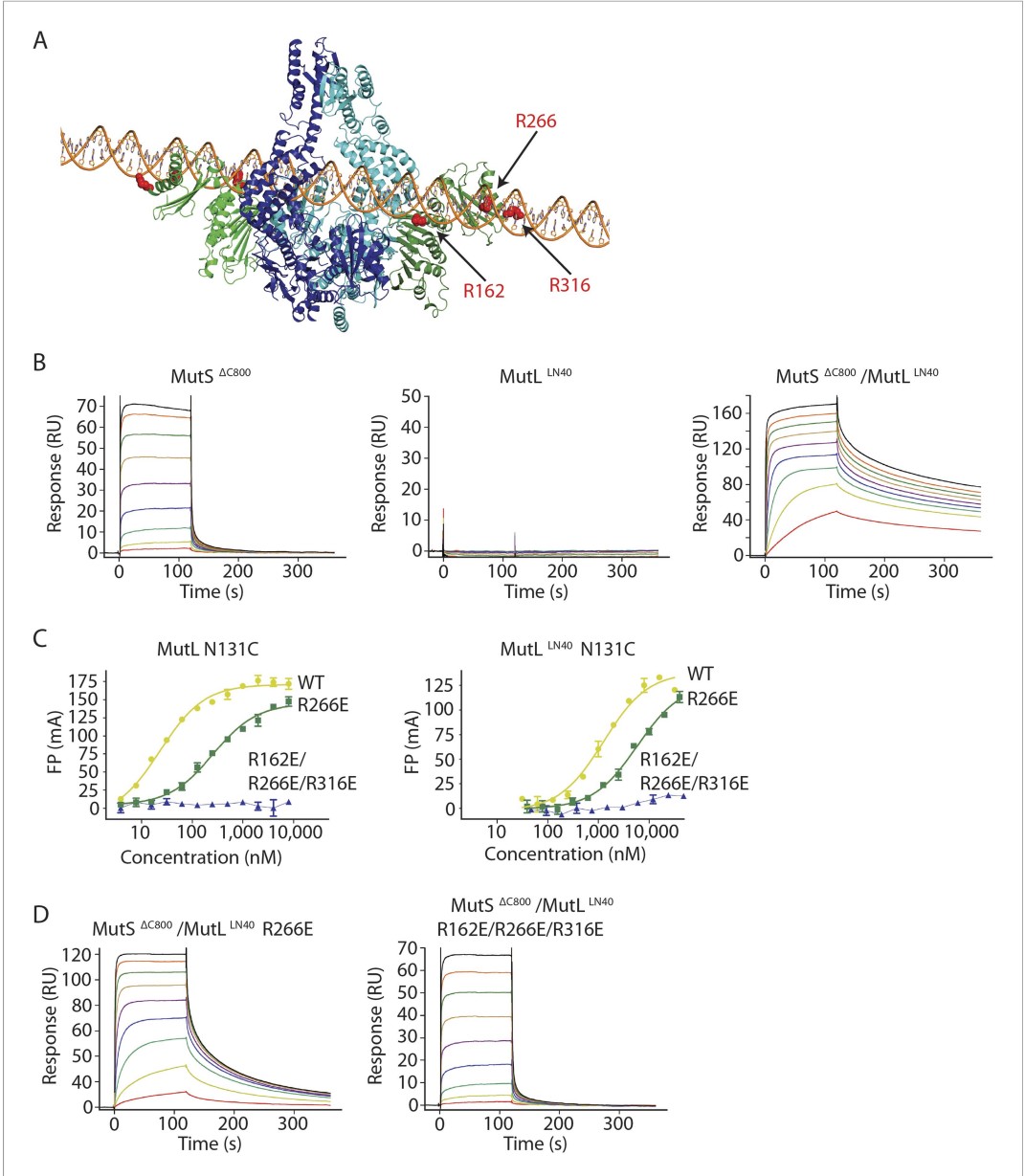

**Figure 4**. The MutS sliding clamp positions MutL onto DNA. (**A**) Model of DNA binding by the MutS$^{\Delta C800}$/MutL$^{LN40}$ complex. Three arginines in the MutL$^{LN40}$ DNA-binding groove are shown as red spheres. (**B**) In the presence of ATP, MutS$^{\Delta C800}$ has a fast off-rate from 100-bp DNA and MutL$^{LN40}$ alone does not bind DNA under physiological salt (150 mM KCl), while the crosslinked MutS$^{\Delta C800}$/MutL$^{LN40}$ complex releases slowly from DNA. (**C,D**) Mutations in the DNA-binding groove of MutL reduce its DNA-binding ability (observed in low salt, 50 mM KCl) (**C**) and affect release rates of the MutS$^{\Delta C800}$/MutL$^{LN40}$ complex in physiological salt conditions (**D**).

The following figure supplement is available for figure 4:

**Figure supplement 1**. DNA binding by the MutS$^{\Delta C800}$/MutL$^{LN40}$ complex.

with the same sequence. This shows that the kinking is concurrent with DNA binding by MutS$^{\Delta C800}$, while kinking is not observed when homoduplex is used (*Figure 5A*, *Figure 5—figure supplement 1A*). When the assay is performed in the presence of ATP, MutS$^{\Delta C800}$ binds and kinks the DNA but subsequently releases due to sliding clamp formation, after which an equilibrium is reached between rebinding and release (*Figure 5B*, *Figure 5—figure supplement 1B*).

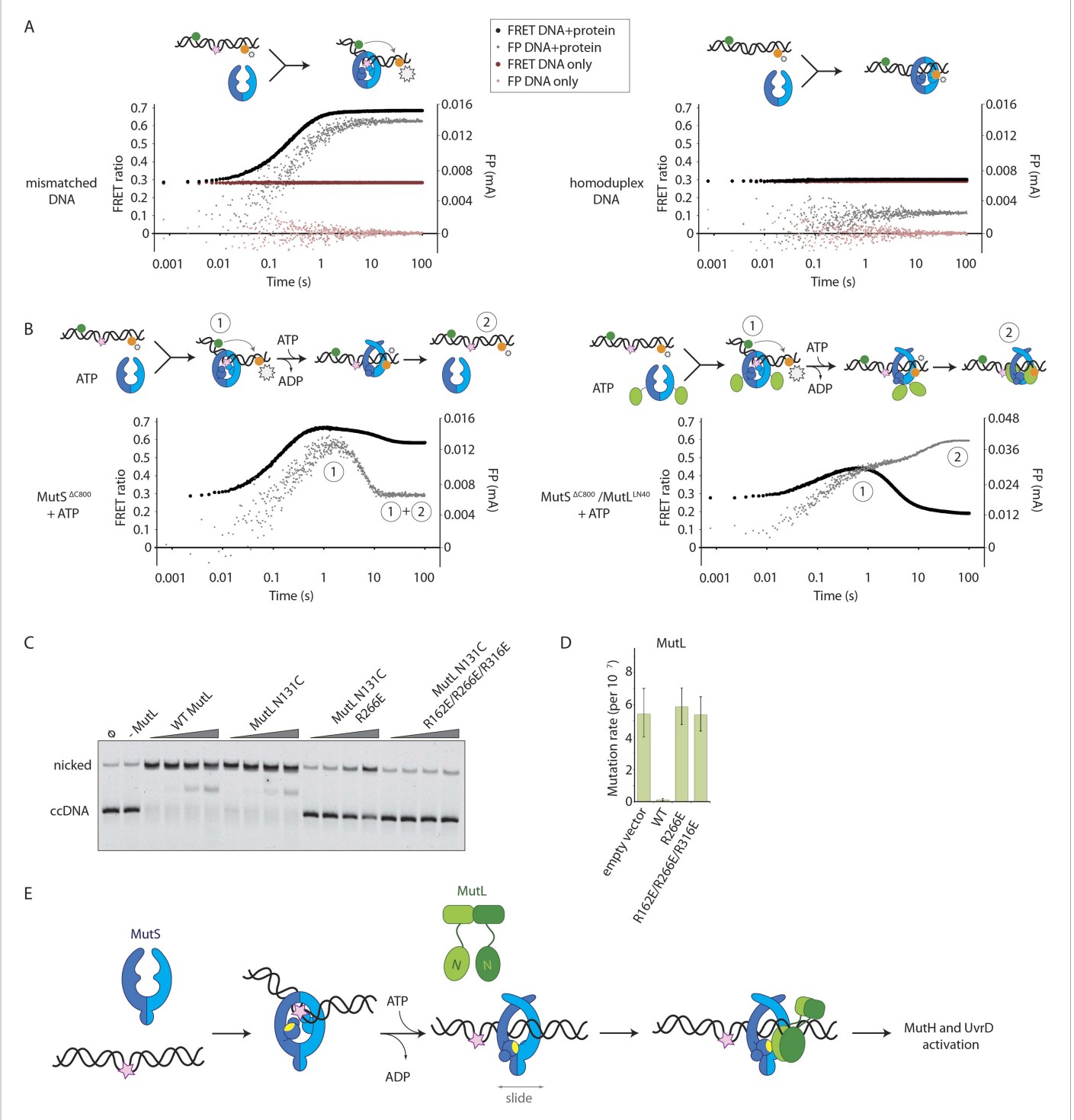

**Figure 5**. Implications for DNA mismatch repair initiation. (**A**) Stopped-flow FRET and FP assay shows kinking of 45-bp DNA by MutS$^{\Delta C800}$ binding only if there is a mismatch. Magnitude of FRET events are indicated by stars in the cartoon. (**B**) While MutS$^{\Delta C800}$ initially kinks the DNA and subsequently releases in the presence of ATP, the MutS$^{\Delta C800}$/MutL$^{LN40}$ shows a secondary FP event without kinking the DNA. (**C**) Nicking assay of mismatch containing closed circular DNA (ccDNA) shows that WT or single-cysteine MutL can activate MutH, while mutations in the DNA-binding groove of MutL strongly impair the activation. (**D**) Spontaneous mutation rates after complementing MutL-deficient cells shows that the DNA-binding ability of MutL is essential for MMR in vivo. Error bars represent 95% confidence intervals. (**E**) Model for MMR initiation. After MutS undergoes an ATP-induced conformational change to allow binding of both subunits to one MutL molecule, MutL N-termini can interact and possibly dimerize, to be loaded onto DNA where MutL can activate downstream effectors.

*Figure 5. continued on next page*

*Figure 5. Continued*

The following figure supplement is available for figure 5:

**Figure supplement 1**. DNA kinking by MutS$^{\Delta C800}$ and MutS$^{\Delta C800}$/MutL$^{LN40}$.

In the presence of the crosslinked complex we observed a two-step sequence of events (*Figure 5B*). The first increase in FP is consistent with mismatch recognition by MutS$^{\Delta C800}$, simultaneous with an increase in FRET due to kinking of the DNA. A second event increases FP even more but reduces the FRET signal to below starting value (*Figure 5—figure supplement 1B*). This can be explained by release of the mismatch (unkinking) and sliding clamp formation. Now, however, the complex does not slide off the DNA but instead the MutL$^{LN40}$ is docked onto the DNA to keep the complex bound, as observed in the SPR assays (*Figure 4B*) and by the increase in FP (*Figure 5B*). At this time, since DNA has been pushed to the new channel, it is not kinked any more but kept relatively rigid by the MutL$^{LN40}$ binding. This, and interaction of the fluorophore itself with bound protein, can explain the lowered FRET. A similar straightening of DNA relative to the unbound DNA was previously observed upon ATP-dependent MutS release in SAXS experiments using DNA labelled with gold-clusters (*Hura et al., 2013a*). The effect is also present to lesser extent when using a mixture of MutS$^{\Delta C800}$ with WT MutL in this setup (*Figure 5—figure supplement 1C*). The result indicates that MutL$^{LN40}$ loading occurs after mismatch recognition and sliding clamp formation by MutS$^{\Delta C800}$.

Since we observed that upon sliding clamp formation, MutS loads MutL onto DNA, we wondered whether this DNA loading step is essential for MMR. Indeed we observed a correlation with the DNA binding ability of MutL for MutH activation (*Figure 5C*). Moreover, the DNA-binding mutants of MutL impair in vivo MMR (*Robertson et al., 2006*) (*Figure 5D, Table 2*), indicating that loading of MutL onto DNA after mismatch recognition is essential for MMR.

## Discussion

Taken together, our data reveal how the large conformational changes within MutS after mismatch recognition promote MMR activation. In the mismatch and ATP activated state MutS pushes DNA into a new channel, which allows sliding of the protein over DNA. The new state with the clamps crossed over the DNA explains the stability of the MutS sliding clamp on DNA (*Schofield et al., 2001*; *Lebbink et al., 2010*; *Jeong et al., 2011*), as electrostatic interactions between DNA and the positive charges lining the new channel may stabilize the new clamp conformation. The conformational change pushes the connector domain away from the center and on top of the ATPase domains, to provide a second interface for the MutL protein that binds to the opposing MutS subunit, while DNA in the new MutS channel can also contribute to MutL binding. This loads the N-terminal domains of MutL onto the DNA and the MutL binding delays the sliding of MutS (*Figure 5E, Video 2*). The loading step of MutL onto DNA is required for MutH activation and nicking (*Figure 5C*) (*Junop et al., 2003*; *Robertson et al., 2006*), while UvrD loading and activation at this nick (*Yamaguchi et al., 1998*) would follow similar validation. In this way, the requirement of the MutS conformational change for full MutL interaction is a sophisticated validation mechanism, which presumably is conserved in the eukaryotic homologs. It ensures that repair is only initiated when necessary, and due to the MMR system DNA replication can be completed with few errors incorporated in the genome.

The complete transition from mismatch binding to sliding clamp state is likely to take multiple steps (*Qiu et al., 2012*). First a single ATP will bind, leading to a stabilized asymmetric nucleotide state of MutS on the mismatch (*Antony and Hingorani, 2004*; *Antony et al., 2006*; *Monti et al., 2011*), followed by binding of the second ATP (*Mazur et al., 2006*; *Hargreaves et al., 2010*). Meanwhile MutS will undergo two separate ATP-induced events, the tilting of the subunits that push DNA into a new channel and the rearrangement of the connector domain (and the associated mismatch binding domain) that together generate a new MutL interface.

These two movements could potentially be uncoupled. MutS binding to a non-hydrolysable ATP analog can already cause a closed clamp-like state, (i.e. perform the tilting movement) as supported by SAXS analysis (*Hura et al., 2013b*), but may possibly not change the conformation of the mismatched binding domain (*Qiu et al., 2012*), as consequence of the connector domain movement. This would explain how MutS with ATPγS (or with ATP for a mutant that cannot hydrolyse nucleotides

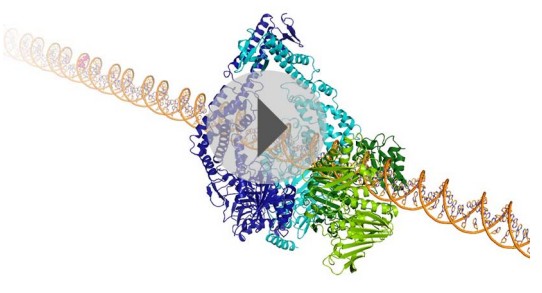

**Video 2.** Model for initiation of DNA mismatch repair. After MutS (cyan/blue) has recognized a mismatch in DNA (in orange; mismatch shown as pink spheres), it will bind ATP which triggers a conformational change in which the subunits tilt across each other and the connector domains move outward. This pushes the DNA to a new channel, where MutS fits as a loose ring around the DNA duplex and can behave as a sliding clamp. The N-terminal domain of MutL (green) can specifically recognize this state by binding two interfaces simultaneously. This loads MutL onto the DNA, where the N-terminal domains could dimerize and downstream effectors can be activated.

[E694A] [*Jacobs-Palmer and Hingorani, 2007*]) could form a closed clamp state that can no longer be loaded onto DNA (*Gradia et al., 1999*; *Jacobs-Palmer and Hingorani, 2007*; *Cristóvão et al., 2012*), but nevertheless is not sufficient to bind MutL.

Our data do not address the order of the two events, tilting and connector movement, or how they relate to the two ATP binding events. Observed conformational changes resulting in ternary complex and sliding clamp formation have previously been suggested to be independent (*Mendillo et al., 2010*). Indeed our structure does suggest that rearrangement of a single connector domain (in the subunit equivalent to the 'MSH2' subunit; (*Mendillo et al., 2009*) is sufficient for the complex formation with MutLα (*Hess et al., 2006*; *Hargreaves et al., 2010*, *2012*). This might allow MSH6 to initially remain bound to the mismatch, consistent with models that consider transient asymmetric nucleotide states involved in mismatch verification and possibly ternary complex formation (*Antony and Hingorani, 2004*; *Hess et al., 2006*; *Lebbink et al., 2006*; *Mazur et al., 2006*; *Hargreaves et al., 2010*; *Monti et al., 2011*; *Qiu et al., 2012*). Another question that is unclear is where the loading of MutL onto DNA takes place. It could occur on or close to the mismatch itself, but it is also possible that MutS first slides before loading MutL on DNA.

Once the sliding clamp conformation is reached, the complex no longer interacts with the mismatch (*Gorman et al., 2012*). The clamp state loads MutL onto DNA, stabilizes a straight form of the DNA (*Figure 5B*) (*Hura et al., 2013a*) and triggers the conformational changes of MutS. These involve movements in the C-terminal domains (*Guarné et al., 2004*) to form a ring around the DNA and ATP binding by the N-terminal domains of MutL to generate the state that activates MutH and UvrD (*Prolla et al., 1994*; *Drotschmann et al., 1998*; *Ban et al., 1999*; *Acharya et al., 2003*).

In conclusion, we have used single-cysteine mutants and chemical crosslinking to trap and analyze a relevant MMR intermediate state that has long been elusive. This sliding clamp state of MutS bound to a MutL domain is highly informative. It corresponds to a reaction intermediate that occurs during a series of conformational changes triggered by mismatch recognition, and explains why specifically this conformation of MutS is able to recruit MutL. The presented combination of structural and biophysical methods provides a powerful approach to resolve conformational changes within large and transient protein complexes that form and act during biologically relevant processes.

## Materials and methods

### Proteins

MutS mutants were created in the *mutS* gene in vector pET-3D (*Lamers et al., 2000*; *Giron-Monzon et al., 2004*; *Manelyte et al., 2006*; *Winkler et al., 2011*) or vector pET15b (*Manelyte et al., 2006*; *Winkler et al., 2011*) (for His-tagged MutS constructs in FRET assays). MutL mutants were generated in the *mutL* gene in plasmid pTX418 (*Feng and Winkler, 1995*; *Ban and Yang, 1998*). Single-cysteine MutS and MutL constructs were obtained as described (*Giron-Monzon et al., 2004*; *Groothuizen et al., 2013*). Mutant and WT MutS and MutL proteins were purified as described previously (*Lamers et al., 2000*; *Manelyte et al., 2006*), except that in the buffers KCl was used instead of NaCl (final gel filtration buffer for MutS: 25 mM Hepes pH 7.5, 150 mM KCl, 1 mM DTT; for MutL: 20 mM Tris pH 8.0, 0.5 M KCl, 10% glycerol, 1 mM DTT).

MutH was purified as follows: *E. coli* BL21(DE3) cells were transformed with MutH expression plasmid pTX417 (*Feng and Winkler, 1995*) and plated onto LB agar with 50 µg/ml carbenicillin. A colony was picked and cells were grown in LB with 50 µg/ml carbenicillin at 37°C to OD600 ~0.6 and induced with 1 mM isopropyl 1-thio-β-D-galactopyranoside for 4 hr. Cells were harvested and resuspended in binding buffer (25 mM Tris pH 8.0, 300 mM KCl, 10 mM imidazole, 0.2 mM DTT) with 1 mM PMSF and protease inhibitors (Roche Diagnostics, F. Hoffmann-La Roche Ltd, Switzerland) and lysed by sonication. The cleared supernatant was incubated with Talon resin (Clonetech Laboratories, Takara holdings inc, Japan) for 30 min on ice. Beads were washed using binding buffer with 1 M KCl, and MutH was eluted with 250 mM imidazole in binding buffer. The His-tag was removed by cleavage with Thrombin protease (~5 units thrombin/mg MutH; GE Healthcare, Fairfield, California) while dialyzing against 20 mM Tris pH 8.0, 100 mM KCl, 0.2 mM DTT for 2 hr at 22°C followed by overnight incubation at 4°C. The mixture was brought to 20 mM imidazole, incubated with Talon beads to remove uncleaved protein, and loaded onto a heparin column equilibrated in buffer A (25 mM Tris pH 8.0, 0.1 M KCl, 1 mM DTT). MutH was eluted using a gradient of 0.1–1.0 M KCl in buffer A, pooled and diluted twofold with buffer A and loaded onto a MonoQ column equilibrated with buffer A. MutH was eluted using the same gradient, pooled and dialyzed overnight against 25 mM MES pH 5.5, 150 mM KCl, 1 mM DTT. MutH was loaded onto a MonoS column equilibrated with 25 mM MES pH 5.5, 0.1 M KCl, 1 mM DTT and eluted using a 0.1–1.0 M KCl gradient. Peak fractions were pooled, concentrated using Centriprep 10 and loaded onto a Superdex 75 column equilibrated with 25 mM Tris pH 8.0, 250 mM KCl, 1 mM DTT. Peak fractions were pooled, concentrated, flash frozen in 25 mM Tris pH 8.0, 250 mM KCl, 1 mM DTT, 50% glycerol and stored at −80°C.

## Small-scale protein crosslinking

Single cysteine MutS$^{\Delta C800}$ and His-tagged MutL$^{LN40}$ proteins were reduced with 10 mM DTT for 20 min and O/N dialyzed into buffer B (25 mM Hepes pH 7.5, 400 mM KCl, 5 mM MgCl$_2$, 10% glycerol) at 4°C, to remove DTT. MutS$^{\Delta C800}$ (0.57 µM) was incubated with 100-bp DNA containing a G:T mismatch (AAACAGGCTTAGGCTGGAGCTGAAGCTTAGCTTAGGATCATCGAGGATC<u>G</u>AGCTC GGTGCAATTCAGCGGTACCCAATTCGCCCTATAGGCATCCAGGTT annealed with AACCTGGAT GCCTATAGGGCGAATTGGGTACCGCTGAATTGCACCGAGCT<u>T</u>GATCCTCGATGATCCTAAGCTAAG CTTCAGCTCCAGCCTAAGCCTGTTT, 0.29 µM) for 25 min on ice in buffer C (25 mM Hepes pH 7.5, 125 mM KCl, 5 mM MgCl$_2$). MutL$^{LN40}$ (4 µM) was incubated with 5 mM ATP for 25 min on ice. MutS$^{\Delta C800}$/DNA and MutL$^{LN40}$/ATP samples were then combined to final protein concentrations 0.4 µM (DNA concentration 0.2 µM) and additional ATP was added to a final concentration of 1 mM. Samples were then incubated for 10 min at RT, after which they were adjusted to 37°C for 2 min. Crosslinker (BMOE or BM[PEO]$_3$, Pierce, Thermo Fisher scientific, Waltham, MA, dissolved to 0.5 mM in DMSO) was added to a final concentration of 50 µM and samples were incubated for exactly 2 min at 37°C. Reactions were stopped by adding protein loading buffer with DTT and crosslinking was assessed on SDS-PAGE gels stained with coomassie.

## MutS$^{\Delta C800}$/MutL$^{LN40}$ complex purification

To obtain crystallizable amounts of crosslinked MutS$^{\Delta C800}$/MutL$^{LN40}$ complex, equimolar amounts of MutS$^{\Delta C800}$ D246C and His-tagged MutL$^{LN40}$ N131C (or with additional arginine mutations) were reduced and dialyzed separately, as described above. MutL$^{LN40}$ was diluted to 2 µM in buffer D (25 mM Hepes pH 7.5, 400 mM KCl, 10% glycerol) and incubated with a 5-fold molar excess of BM (PEO)$_3$ (from 50 mM solution in DMSO) for 10 min at 4°C. The low MutL$^{LN40}$ concentration prevented the formation of MutL$^{LN40}$-MutL$^{LN40}$ crosslinks, while the excess crosslinker ensured each MutL$^{LN40}$ to react with one maleimid group so that the other reactive side of the crosslinker remained available. The MutL$^{LN40}$ was then bound to Talon beads and the beads were subsequently washed with 20 column volumes of buffer D and 20 column volumes of buffer E (25 mM Hepes pH 7.5, 150 mM KCl, 10% glycerol, 5 mM imidazole) to remove excess crosslinker. MutS$^{\Delta C800}$ was incubated for 10 min with equimolar amounts of 30-bp DNA with a G:T mismatch at position 9 (AGCTGCCA<u>G</u>GCACCAGTGT CAGCGTCCTAT annealed with ATAGGACGCTGACACTGGTGC<u>T</u>TGGCAGCT) in buffer C. The DNA-bound MutS$^{\Delta C800}$ was then added to the Talon-bound MutL$^{LN40}$, and 30-fold excess ATP was immediately added after which everything was incubated to crosslink for 1 hr at 4°C. The beads were then washed with 10 column volumes buffer E to remove MutS$^{\Delta C800}$-MutS$^{\Delta C800}$ crosslinks, after which

the protein was eluted in buffer E with 300 mM imidazole and DTT was added to quench excess crosslinker. The protein was bound to a 5 ml heparin column (GE Healthcare, Fairfield, California) and eluted with a 0.1–1 M KCl gradient, which removed the DNA from the protein. The elution was subsequently concentrated and purified with size-exclusion chromatography in buffer B containing 1 mM DTT, for which two S200 16/60 columns were coupled resulting in one long column. The MutS$^{\Delta C800}$/MutL$^{LN40}$ protein peak was then concentrated, after which the MutS$^{\Delta C800}$ concentration was estimated using $\varepsilon = 95,238$ and the whole process (including DTT incubation and dialysis) was repeated to obtain S$_2$L$_2$ complexes. The resulting protein (5–10% final yield) was concentrated to 80–90 μM (expressed in MutS monomer concentrations; $\varepsilon = 94,660$) and flash-frozen until further use.

## Crystallization and structure solution

For crystallization, 50 μM MutS$^{\Delta C800}$/MutL$^{LN40}$ complex was incubated with 25 μM DNA containing a G:T mismatch (27-bp: TGCCAGGCACCAGTGTCAGCGTCCTAT annealed with ATAGGACGCTGACACTGGTGCTTGGCA or 100-bp, same sequence as above) for 25 min on ice. AMP-PNP was subsequently added to a concentration of 1 mM and the protein was crystallized at 4°C using vapor diffusion in 9–12% PEG-8000, 100 mM Tris pH 7.0, 200 mM MgCl$_2$, and 80–450 mM sodium malonate. Microseeding was used to increase crystal nucleation. Crystals were cryoprotected in mother liquor supplemented with 25% ethylene glycol and 100 mM KCl before flash-cooling in liquid nitrogen. Diffraction data were collected at 100 K at beamline ID-29 at the ESRF or beamline PX-III at the SLS.

Crystallographic data were processed with XDS (*Kabsch, 2010*) or iMOSFLM (*Powell et al., 2013*) and scaled using Aimless from the CCP4 suite (*Winn et al., 2011*). Crystal structures were solved in consecutive steps of finding domains using Phaser (*McCoy et al., 2007*). Several search models were used, but best results were obtained with domains from chain A of PDB entry 1W7A as search models for MutS$^{\Delta C800}$ and chain A from PDB entry 1BKN for MutL$^{LN40}$, while clear density for residues 150–164 of MutL$^{LN40}$ allowed building as in PDB entry 1NHJ. The search process was improved by going back and forth between the different datasets to find missing domains. Initial structure solution was performed starting from crystal form 1 as follows: first, a search model consisting of residues 267–800 of chain A of PDB entry 1W7A (MutS) was searched twice using Phaser, which resulted in a solution with these chains forming a tilted MutS dimer. Next, this solution was used together with a search model consisting of chain A of 1BKN (MutL$^{LN40}$), which placed this protein against the ATPase domain of one MutS subunit. Then, the second MutLLN40 was found with Phaser using a brute rotation search of 15° around the angle that would orient this MutL$^{LN40}$ on the other side of the MutS dimer in a similar manner as the first, and automated translation, packing and refinement steps by Phaser indeed placed the MutL$^{LN40}$ in the symmetrically equivalent position. One connector domain (residues 128–266 of chain A of 1W7A) was then found with Phaser, and the second connector domain was placed using similar steps as for the second MutL$^{LN40}$ search. Thus the search identified the equivalent dimeric counterpart three times for separate parts of the complex (the main MutS chain, MutL$^{LN40}$ and connector domains). The resulting MutS-MutL$^{LN40}$ complex structure could then be used as a search model in all crystal forms and easily identified equivalent complexes in each of those (present three times in the asymmetric units in the 6.6 Å and 7.6 Å datasets). Finally, for crystal form 1, an additional 'half complex' was found with Phaser using one MutS chain and one MutL$^{LN40}$ chain of the existing complex structure. This second complex forms a symmetry-generated dimer over a twofold axis, with similar MutS-MutL$^{LN40}$ interfaces, but the MutS clamp domains in this crystallographic dimer could not be modeled. This second conformer forms a more compact MutS dimer, probably due to crystal packing, but since it has identical interfaces with MutL$^{LN40}$ we focussed on the main conformation throughout this paper. Excellent quality of the structure solutions after molecular replacement with the complete but unrefined models is evident from the Phaser statistics: TFZ = 9.0/LLG = 996 for 4.7 Å; TFZ = 14.2/LLG = 899 for 6.6 Å; and TFZ = 13.0/LLG = 795 for the 7.6 Å dataset.

Refinement was first performed using rigid body refinement in REFMAC5 (*Murshudov et al., 1997, 2011*), for which the following domains of MutS were defined: residues 128–266, 267–765, 766–800; and for MutL: residues 20–204, 205–331. Next, limited restrained refinements were performed, first using ProSMART-generated external restraints (*Nicholls et al., 2012*) to the PDB_REDO-optimized (*Joosten et al., 2012*) entries of chain A of 1W7A and chain A of 1BKN in order to ensure consistency with prior observations, followed by TLS and jelly-body refinement in latter stages. PDB_REDO-optimized homologues were used for external restraint generation in order to

maximize reliability of the prior structural information. All refinements were performed using REFMAC5 (*Murshudov et al., 1997*, *2011*). During refinement, clear density became visible for missing residues 150–164 of the MutL$^{LN40}$ subunits, which followed the conformation of PDB entry 1NHJ. Interestingly, this conformation was different from that in the MutL search model state, indicating this to be real signal, and not due to bias from the search model. Also, AMP-PNP could be placed in density in the nucleotide binding sites of MutS. During intermediate stages, PDB_REDO and MolProbity (*Chen et al., 2010*) were used to correct geometry and perform side-chain flips. After refinement, all structures were in the 97th–100th Clashscore and 98th–100th MolProbity score percentiles. Refinement and data collection statistics can be found in *Table 1*. Figures and videos were generated with MacPyMOL (http://www.pymol.org), interpolations between conformations were created with LSQMAN (*Kleywegt and Jones, 1994*) and electrostatic surface with CCP4mg (*Winn et al., 2011*). Protein interface areas were calculated using PISA (*Winn et al., 2011*) for which the missing loop of residues 126–131 of MutL$^{LN40}$ in interface 2 was modeled as in PDB entry 1NHJ.

## MutS conformational changes

To look at changes within MutS dimers, we used MutS D835R dimer (*Manelyte et al., 2006*; *Groothuizen et al., 2013*) variants that do not form tetramers, with single cysteines in positions R449C (His-tagged), D246C, S798C, or A336C. The proteins were labeled with Alexa Fluor 488 or Alexa Fluor 594 maleimide (Invitrogen, Thermo Fisher scientific, Waltham, MA) according to the manufacturers instruction. Excessive dye was removed using Zeba Spin Desalting columns (Thermo Fisher scientific, Waltham, MA) and the degree of labeling determined from the absorbance spectra recorded from 220–700 nm (nanodrop) according to the manufactures instructions.

Clamp-domain crossover movement and connector domain movement within MutS dimers were measured using FRET in which fluorescence emission spectra were recorded with excitation at either 485 nm (5 nm slit width) for FRET or 590 nm (5 nm slit width) for direct acceptor measurements. FRET was determined by the ratio between signal at 485 and 615 nm while direct acceptor was determined by the ratio between signal at 590 and 615 nm and after correction for spectral crosstalk the ratio FRET/acceptor was calculated, and normalized for unbound protein. Heterodimers of single-cysteine MutS variants labeled with Alexa Fluor 488 and Alexa Fluor 594, respectively, were allowed to form by mixing 200 nM of each protein and incubation at 22°C for at least 30 min in the absence of ADP in buffer F (25 mM Hepes pH 7.2, 150 mM KCl and 5 mM MgCl$_2$) supplemented with 0.05% TWEEN-20. Next, 200 nM of 59-bp DNA with a G:T mismatch (TGAAGCTTAGCTTAGGATCATCGAGGATCG AGCTCGGTGCAATTCAGCGGTACCCAATT annealed with AATTGGGTACCGCTGAATTGCACCGA GCTTGATCCTCGATGATCCTAAGCTAAGCTTCA, with blocked ends as described above) was added, followed by addition of 1 mM ATP. As a homoduplex control 240 pM λ-DNA (corresponding to 200 nM of the 59 bp blocked Heteroduplex-DNA) was used.

MutS-DNA FRET was measured in a Hitachi Fluorescence spectrofluorimeter F-4500 (Hitachi Ltd, Japan) (Program FL Solutions 2.0). Fluorescence emission spectra (600–700 nm) were recorded with excitation at either 435 nm (5 nm slit width) for FRET or 590 nm (5 nm slit width) for direct acceptor measurements. FRET was determined by the ratio between signal at 435 and 615 nm while direct acceptor was determined by the ratio between signal at 590 and 615 nm and after correction for spectral crosstalk the ratio FRET/acceptor was calculated. We used 30-bp DNA with or without a G:T mismatch (AATTGCACCGAGCTTGATCCTCGATGATCC annealed with complementary strand or GGATCATCGAGGATCGAGCTCGGTGCAATT), where the T-containing strand had 5' and 3' digoxigenin labels so that both DNA ends were blocked with anti-digoxigenin Fab fragments (Roche Diagnostics, F. Hoffmann-La Roche Ltd, Switzerland). 100 nM of the DNA with 6 μM SYTOX Blue (Invitrogen, Thermo Fisher scientific, Waltham, MA) was mixed with 200 nM MutS variants labeled with Alexa Fluor 594 in buffer F, after which ATP was added to 1 mM to induce the conformational change in MutS.

## In vivo MMR complementation

Spontaneous mutation rates were assessed using acquired rifampicin resistance. Strains KR1517 (*mutS*, as in [*Lamers et al., 2004*]) or GM4250 (gift from M Marinus, [*Aronshtam and Marinus, 1996*]) (*mutL*) were transformed with empty vector or plasmid containing WT or mutant MutS or His-MutL

genes, and plated on LB/agar plates with 50 µg/ml carbenicillin and 30 µg/ml kanamycin. After overnight incubation at 37˚C, single colonies were picked and grown in 10 ml LB with antibiotics to $OD_{600}$ ~1.0. Next, $0.33 \times 10^8$ or $1 \times 10^8$ cells were plated on LB plates with antibiotics and 0.1 mg/ml rifampicin. Plates were O/N incubated at 37˚C and rifampicin resistant colonies were counted. Mutation rates and 95% confidence intervals were determined using Fluctuation AnaLysis CalculatOR with the MSS maximum-likelihood method (http://www.mitochondria.org/protocols/FALCOR.html).

## DNA binding kinetics

SPR experiments for binding $MutS^{\Delta C800}$ D246C or crosslinked $MutS^{\Delta C800}/MutL^{LN40}$ complex to DNA were performed in a Biacore T200 system (GE Healthcare, Fairfield, CA) as described (*Groothuizen et al., 2013*). The experiments were performed in SPR buffer containing 25 mM Hepes pH 7.5, 150 mM KCl, 5 mM $MgCl_2$, 1 mM DTT, 0.05% TWEEN-20 and 1 mM ATP, using immobilized biotinylated 100-bp DNA (same sequence as above) with a fluorescein moiety at the other end.

## MutL-MutS binding assay

Full-length $His_6$-MutL binding to the full-length MutS sliding clamp was assessed using a two-step SPR assay. The resulting graphs are not strictly affinity curves, as changes in MutS stability on DNA contribute to the observed response, but serve to assess the effect of mutations. The SPR buffer was supplemented with 20% glycerol to ensure MutL stability. Before every measurement, anti-fluorescein Fab fragment (Invitrogen, Thermo Fisher scientific, Waltham, MA) was injected to block the fluorescein-labeled DNA (100 bp, see above) ends. MutS sliding clamps were captured on the end-blocked DNA by injecting 200 nM WT or mutant MutS protein (in buffer with 1 mM ATP) for 120 s. Then WT or mutant MutL protein (in buffer with 1 mM ATP) was injected for 120 s, followed by dissociation with buffer only. This was repeated with varying concentrations of MutL.

## DNA binding by MutL

Fluorescence polarization measurements to assess DNA-binding of $MutL^{LN40}$ mutants were performed in low-salt FP buffer with 25 mM Hepes pH 7.5, 50 mM KCl, 5 mM $MgCl_2$, 1 mM DTT and 0.05% TWEEN-20. For full length MutL, the buffer was supplemented with 10% glycerol. A concentration of 0.5 nM of 5′ labeled TAMRA-41-bp DNA (ATAGGACGCTGACACTGGTGCTTGGCAGCTTCTAATTCGAT annealed with complementary strand) was used. MutL proteins were serial diluted in black 96-well microplates (PerkinElmer Inc, Waltham, MA) in 100 µl volumes. Polarization of the TAMRA label was read out in a PHERAstar FS machine (BMG Labtech GmbH, Germany) with an 540/590 (excitation/emission) FP module.

## DNA kinking assays

Stopped-flow assays to assess DNA binding and kinking were performed in buffer containing 25 mM Hepes pH 7.5, 150 mM KCl, 5 mM $MgCl_2$, 1 mM DTT, 0.05% TWEEN-20 and 10 µM ADP, with or without 1 mM ATP. One syringe contained 100 nM of 45-bp DNA with or without a G:T mismatch (GTCATCCTCG[T*]CTCAAGCTGCCA<u>G</u>GCACCAGTGTCAGCGTCCTAT annealed with complementary strand or ATAGGACGC[T*]GACACTGGTGC<u>TT</u>GGCAGCTTGAGACGAGGATGAC) which was either labeled with Alexa Fluor 594 at position 11 in the top strand and Alexa Fluor 488 at position 10 in the bottom strand (indicated by T*), or with 5′-labeled with TAMRA in the top strand. The other syringe contained 400 nM $MutS^{\Delta C800}$ D246C or crosslinked $MutS^{\Delta C800}/MutL^{LN40}$ complex. For assays with double-labeled DNA, donor fluorophores were excited at 473 nm and measured using filters 540IB + 540IK, while acceptor fluorophores were measured at the same time using an OG590 filter. For experiments with TAMRA-labeled DNA, the fluorophore was excited at 545 nm and OG540 filters were used for read-out. Samples were co-injected in a KinetAsyst SF-61DX2 stopped-flow machine (TgK Scientific, UK) fitted with R10699 photomultiplier tubes (Hamamatsu Photonics K.K., Japan) at 15˚C and measured for 100 s, which was repeated 5–10 times and averages were calculated.

## MutH activation assay

Circular DNA containing a single G:T mismatch and 12 hemi-methylated GATC sites was prepared via primer extension on single stranded DNA from a derivative of pGEM-13Zf (gift from J Jiricny) as

described (*Baerenfaller et al., 2006*) with the exception that closed circular DNA was purified from gel using a Wizard gel purification kit (Promega Corporation, Madison, WI). To enable quantification, an Alexa Fluor 647 labeled oligo (IBA GmbH, Germany) was used: CCAGACGTCTGTC<u>G</u>ACGTTGG-GAAGCT[T*]GAGTATTCTATAGTGTCACCT, where the <u>G</u> is nucleotide forming a G:T mismatch and the T* is the Alexa Fluor 647 labeled nucleotide. Nicking reactions contained 25 mM Hepes KOH pH 7.5, 150 mM KCl, 0.1 mg/ml BSA, 5 mM $MgCl_2$, 1 mM DTT, 1 mM ATP, 0.5 nM circular DNA, 200 nM MutS, 200 nM WT MutL, single-cysteine MutL N131C, MutL N131C R266E or MutL N131C R162E/R266E/R316E and 100 nM MutH as well as twofold dilutions thereof. Control reactions contained either no protein or 200 nM MutS and 100 nM MutH. Reactions were incubated for 5 min at 37°C and stopped with an equal volume of 20% glycerol, 1% SDS and 50 mM EDTA. Samples were analyzed on 0.8% agarose gels supplemented with 1 µg/ml ethidium bromide, run in 1x TAE. Conversion of covalently closed circles into nicked product was visualized using the fluorescence of the Alexa Fluor 647 label using a Typhoon Trio Imager (GE Healthcare, Fairfield, CA) with excitation at 633 nm and emission filter 670BP30.

## ATPase assay

ATPase activity of WT MutS and MutS P595A/I597A/M759D was measured by coupling ATP hydrolysis to oxidation of NADH as in (*Lamers et al., 2004*). MutS protein (5 µM) was mixed with 3.125–500 µM ATP and hydrolysis was measured in a spectrophotometer during 5 min.

## MutL^LN40 dimerization assay

Crosslinked MutS^ΔC800/MutL^LN40 complex (1 mg/ml) was incubated for 5 min on ice with equimolar amounts of 100-bp DNA containing a G:T mismatch (sequence as in main text). MutL^LN40 (2 mg/ml) was incubated with the MutS^ΔC800/MutL^LN40/DNA complex or with DNA only, and 1 mM AMP-PNP as described (*Ban and Yang, 1998*). Samples were injected onto a S200 5/150 column in buffer containing 20 mM Tris pH 8.0, 150 mM KCl, 0.1 mM EDTA, 5 mM $MgCl_2$, 1 mM DTT and 5% glycerol. Eluted fractions were analyzed on SDS-PAGE stained with coomassie.

## Acknowledgements

We are grateful for contributions of group members, to Randy Read for assistance with molecular replacement, and to Lea Geissert, Miguel Keidel, Gaelle Cyriale Ngatcheu Famou, Rosine Djamfa and Matthias Trohart for assistance with FRET assays. We thank Jacques Neefjes, Hein te Riele and Thijn Brummelkamp for critical reading of the manuscript. Structure coordinates have been deposited in the Protein Data Bank under accession codes 5AKB, 5AKC, 5AKD. This research was funded by European Community's Seventh Framework Programme mismatch2model HEALTH-F4-2008-223545, the Centre for Biomedical Genetics, and NWO-CW ECHO 711.011.011 (to TS) and VIDI 700.58.428 (to JL). The authors declare to have no competing interests.

## Additional information

### Funding

| Funder | Grant reference | Author |
| --- | --- | --- |
| European Commission | mismatch2model HEALTH-F4-2008-223545 | Joyce HG Lebbink, Peter Friedhoff, Titia K Sixma |
| Ministerie van Onderwijs, Cultuur en Wetenschappen | Center for Biomedical Genetics | Titia K Sixma |
| Nederlandse Organisatie voor Wetenschappelijk Onderzoek | CW Echo 711.011.011 | Titia K Sixma |
| Nederlandse Organisatie voor Wetenschappelijk Onderzoek | CW VIDI 700.58.428 | Joyce HG Lebbink |

The funders had no role in study design, data collection and interpretation, or the decision to submit the work for publication.

## Author contributions

FSG, Performed and designed molecular biology, biophysical, structural and in vivo assays, and wrote the manuscript with contributions from TKS, AF, JHGL and PF; IW, Contributed initial MutS/MutL crosslinking; MC, ADM, Set up FRET experiments with labeled MutS and stopped-flow experiments; AF, Contributed and designed SPR and stopped-flow biophysics; HHKW, Contributed to biochemical and in vivo assays; AR, Contributed to initial MutS/MutL crosslinking; NH, Contributed the MutH activation assay; RAN, GNM, Contributed to structure refinement; JHGL, Contributed the MutH activation assay and contributed to supervision of the study; PF, Designed all tools, supervised FRET experiments and contributed to supervision of the study; TKS, Supervised the study with contributions from JHGL and PF

# Additional files

## Major datasets

The following datasets were generated:

| Author(s) | Year | Dataset title | Dataset ID and/or URL | Database, license, and accessibility information |
|---|---|---|---|---|
| Groothuizen FS, Winkler I, Cristovao M, Fish A, Winterwerp HHK, Reumer A, Marx AD, Hermans N, Nicholls RA, Murshudov GN, Lebbink JHG, Friedhoff P, Sixma TK | 2015 | MutS in complex with the N-terminal domain of MutL—crystal form 1 | http://www.rcsb.org/pdb/explore/explore.do?structureId=5akb | Publicly available at the RSCB Protein Data Bank (Accession no 5AKB). |
| Groothuizen FS, Winkler I, Cristovao M, Fish A, Winterwerp HHK, Reumer A, Marx AD, Hermans N, Nicholls RA, Murshudov GN, Lebbink JHG, Friedhoff P, Sixma TK | 2015 | MutS in complex with the N-terminal domain of MutL—crystal form 2 | http://www.rcsb.org/pdb/explore/explore.do?structureId=5akc | Publicly available at the RSCB Protein Data Bank (Accession no 5AKC). |
| Groothuizen FS, Winkler I, Cristovao M, Fish A, Winterwerp HHK, Reumer A, Marx AD, Hermans N, Nicholls RA, Murshudov GN, Lebbink JHG, Friedhoff P, Sixma TK | 2015 | MutS in complex with the N-terminal domain of MutL—crystal form 3 | http://www.rcsb.org/pdb/explore/explore.do?structureId=5akd | Publicly available at the RSCB Protein Data Bank (Accession no 5AKD). |

Standard used to collect data: PDB validation reports are supplied.

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
