## [Decision Letter]

Thank you for sending your work entitled “MutS/MutL crystal structure reveals that the mutS sliding clamp loads MutL onto DNA” for consideration at *eLife*. Your article has been favorably evaluated by John Kuriyan (Senior editor) and three reviewers, including Leemor Joshua-Tor (Reviewing editor), and Randy Read.

The Reviewing editor and the other reviewers discussed their comments before we reached this decision, and the Reviewing editor has assembled the following comments to help you prepare a revised submission.

This is a strong paper describing a thorough study of the implications of the structure of a trapped MutS:MutL complex on the mechanism of DNA mismatch repair. The structural studies accompanied by FRET, crosslinking and binding studies show that MutS is converted to a sliding clamp conformation to load MutL onto DNA. The paper is well written and clear despite covering a lot of material. The crystal structures along with analyses resulting in many new insights provide at least two major breakthroughs: the MutS ATP bound conformation and the requirement of the MutS conformational change for functional MutL interaction: together these provide an elegant regulatory mechanism for commitment to repair. The additional analysis and testing via mutants, gels, FRET and SPR provide substantial added impact. This is likely to be one of the most important sets of structural results for the DNA mismatch repair field in the last 5 years. There are many novel insights that come directly from these structures and associated analyses, such as the ATP-bound MutS conformation and the rearrangement of the connector domain that creates a second interface with MutL to regulate pathway progression. Moreover, this work certainly advances our understanding of macromolecular machines in general.

The starting point for this study is the determination of the structure of trimmed constructs of MutS and MutL that have been trapped in a transient conformation by crosslinking Cys side chains that are sufficiently close only in the sliding clamp conformation. The process of producing enough stable MutS/MutL dimer for crystallization efforts is itself impressive. The structures are without a doubt important and interesting. Three crystal forms were obtained, and all diffract to relatively low resolution with the best being 4.7Å. At this resolution, care must be taken in structure solution and refinement to overcome the poor parameter:data ratio. The validity of the molecular replacement structure solution is pretty much beyond question, as the statistics for the placement of components are convincing and, of much greater importance, the same complex has been assembled more than once in the course of structure solution. The refinement has been carried out with care to avoid over-fitting, as indicated by the low values for *R*_free_ and the presence of unbiased features showing unmodelled loops and expected ligands. In addition, all the important conformational changes and molecular contacts inferred from the structure have been checked by FRET measurements and cross-linking experiments.

As the authors point out: “During refinement, clear density became visible for missing residues 150-164 of the MutL^LN40^ subunits, which followed the conformation of PDB entry 1NHJ. Interestingly, this conformation was different from that in the MutL search model state, indicating this to be real signal, and not due to bias from the search model.” In analyzing the structures, the authors make several good points in relating the current results to other established data. However, as the MMR field has ongoing strong controversies regarding the sliding clamp and MutL interactions, it will be important to place this exciting new data as firmly as possible into the context of other biophysical work.

The reviewers raised several issues that should be addressed in a revised manuscript:

As mentioned, the findings in this paper are substantial and some aspects of the results did not receive a complete discussion in light of other observations in the field. The reviewers discussed the possibility of suggesting a split into two papers, however, this would slow down the process considerably. Instead, since there is no space limitation, the authors are encouraged to expand their Discussion as pointed out below.

Specific comments:

1) According to section 7 of the submission guidelines, crystal structures must be deposited and the PDB validation reports must be provided to the referees. The referees should have a chance to examine these reports before the paper is accepted, and the PDB codes should be given in the text.

2) In the Abstract, the statement about the structure being based on cysteine crosslinking could be misread as indicating that this was a hybrid structure based largely on inferred distances. It would be better to clarify in the Abstract that the desired conformation was trapped by cysteine crosslinking and that crosslinking experiments were also used to validate the molecular interactions inferred from the structure.

3) Several experiments are presented – FRET, binding, crosslinking – however, in most a control with non-mismatched DNA is missing. There are claims such as “dependent on mismatched DNA” that could not be made without these controls. Some of the FRET experiments should also be done with non-hydrolyzable ATP in addition to ATP. For the FRET experiments – the data in supplementary material should include donor and acceptor curves, rather than just final FRET differences.

4) The presentation is clear and concise, but it is disappointing that more of the paper was not devoted to describing the crystal structures plus their relationships to transient states, as together they answer many questions. For example, the packing within the crystal structure, as a potential filament on DNA, is of particular interest to many but left as supplement. An expanded discussion should address the transient states, the specificity and effect of MutL binding to MutS, and the new channel resulting from a novel conformation of MutS that reveals a rearrangement of the subunits in the MutS^ΔC800^ dimer.

5) Importantly, this work establishes the ATP-bound conformation for MutS dimer with a greater motion than expected from prior MutS structures. In this novel MutS conformation the subunits are tilted across each other by ∼30{degree sign} which explains the stability of the MutS sliding clamp on DNA, compared to the mismatch recognition state. More detail on how this compares to the Rad50 ABC ATPAse ATP-bound conformation would be useful to readers, as the use of ABC ATPases in both mismatch and double-strand break repair has sparked considerable interest. E.g. the ABC ATPase signature helix identified from Rad50 conveys ATP-binding to Rad50 conformations, is this similar or different in the MutS changes in concert with ATP and MutL (Williams GJ et al., ABC ATPase signature helices in Rad50 link nucleotide state to Mre11 interface for DNA repair. Nat Struct Mol Biol. 2011 Apr;18(4):423-31)? Such conformational switching in Rad50 controls pathway progression and the choice between DNA end joining and excision, which may have parallels to these new MutS results or not (Deshpande RA et al., ATP-driven Rad50 conformations regulate DNA tethering, end resection, and ATM checkpoint signaling. EMBO J. 2014 33, 482-500).

6) The authors' state that the “… sliding clamp state of MutS bound to a MutL domain is highly informative.” However their point that “it corresponds to a reaction intermediate that occurs during a series of conformational changes triggered by mismatch recognition” depends upon putting this work firmly into the context of existing results. Given the novel conformation of MutS and low-resolution coupled to issues from the heavy reliance on Cys mutants, cross-linked and truncated constructs and the absent electron density for the DNA, the authors would do well to better relate this work to other recent biophysical results where feasible, e.g. Gorman J, et al., Single-molecule imaging reveals target-search mechanisms during DNA mismatch repair. Proc Natl Acad Sci U S A. 2012 Nov 6;109(45):E3074-83 and DeRocco VC, Sass LE, Qiu R, Weninger KR, Erie DA. Dynamics of MutS-mismatched DNA complexes are predictive of their repair phenotypes. Biochemistry. 2014 Apr 1;53(12):2043-52.

7) The authors also point out that their presented combination of structural and biophysical methods provides a powerful approach to resolve conformational changes within large and transient protein complexes. Thus, it would be useful to relate this model to recent data on the comprehensive conformational states for MutS and the model for bending then straightening steps in MutS damage recognition. X-ray scattering data was interpreted to suggest that nucleotide binding drives MutSβ conformational changes (Hura GL et al., Comprehensive macromolecular conformations mapped by quantitative SAXS analyses. Nat Methods. 2013 Jun;10(6):453-4.) These new observations would seem to be in general agreement with the SAXS results, e.g. “As this is measured in the absence of MutL it suggests that after mismatch binding, ATP is sufficient to induce movement of the connector domain away from the mismatch-recognition position.”

8) For the crosslinked complex, the authors observe a two-step sequence of events, a first increase in FP simultaneous with an increase in FRET due to kinking of the DNA, then a second event increases FP even more but reduces the FRET signal to below starting value in which “… the DNA is not kinked any more but kept relatively rigid by the MutL” – these data seem to support and extend results from X-ray scattering, which showed DNA bending then straightening upon formation of MutS/MutL complexes (Hura GL et al., DNA conformations in mismatch repair probed in solution by X-ray scattering from gold nanocrystals. Proc Natl Acad Sci U S A. 2013 Oct 22;110(43):17308-13).

9) They do not observe the DNA in these structures. They claim that either the DNA is disordered (sliding along the channel that is formed) or that it was released. Could this be experimentally examined? Though not complete verification, were the crystals examined to see whether DNA was in the crystals? In one of the crystal forms the channels of at least 3 copies are aligned – if the DNA is in there – it might show fiber scattering along that axis. Do they see this? Would the DNA be able to stack along there (due to length)? Though they may not have a 2:2:1 complex anymore.

10) In the Introduction, around the third paragraph, it would be good to help out the uninitiated reader by noting that the newly replicated strand is recognised by the lack of adenine methylation.

11) In the first paragraph of the Results, it would be good to mention the nature and length of the crosslinker. Until a clarification in the subsection “MutS sliding clamp recognition by MutL”, it could appear that the authors were referring to disulphide bridge formation.

12) In the second paragraph of the Results, the phrase “purification to obtain the protein in two successive cycles” should be clarified. Does this means that two purification steps were carried out?

13) In the second paragraph of the subsection “MutS sliding clamp recognition by MutL”, it is not really appropriate to compare the 15.5Å distance between C-alpha atoms of Cys residues with the 18Å distance (really 17.8Å according to the Pierce information) between C atoms in the maleimide ring. In the cross-linked state, there will be three additional bonds on each side (CA-CB-SG-C), giving a significantly larger maximum distance between C-alpha atoms.

14) In the legend to Figure 2—figure supplement 1, there should be a capital “D” in the definition of the difference map coefficients, i.e. “mFo-DFc”. The authors should always indicate what types of electron density maps are presented (supplementary figures).

[Editors' note: further revisions were requested prior to acceptance, as described below.]

Thank you for resubmitting your work entitled “MutS/MutL crystal structure reveals that the mutS sliding clamp loads MutL onto DNA” for further consideration at *eLife*. Your revised article has been favorably evaluated by John Kuriyan (Senior editor), a Reviewing editor, and two reviewers. The manuscript has been improved significantly but there are some remaining issues that need to be addressed before acceptance, which should be expedient at that point. These are outlined below.

Overall this is an impressive and high impact manuscript on the MutS-MutL structure. The large and functionally-important conformational change seen in MutS with MutL and nucleotide binding fills a big gap in our understanding with likely broad impacts as noted below. Also for the most part, the authors have addressed the reviewers' comments. However, there are some responses where the authors should take another look. Given the importance of this work, in order to more clearly position it in the field, and improve the impact this paper would have, the authors should consider the following 5 points before publication.

1) In response to point 5 about how this new work related to the Rad50 ABC ATPase family member (specifically recent NSMB and EMBO papers), they mention that while “the signature helix itself is indeed important … and the new position of the connector domain is overlapping with the position of the coiled coil regions in Rad50 in the AMP-PNP bound state” the “detailed interactions driving conformational changes seem different” (i.e. salt-bridge switches are not conserved).

In the text they note similarities to other ABC ATPases, and mention the signature helix, although they do not cite the recent NSMB or EMBO papers that experimentally dissected the role of residues on the signature helix (they rather cite a 2003 review). And in the Discussion they don't discuss whether or not a different set of salt bridge switches may coordinate conformational changes. To be fair this may be complicated by the low-resolution nature of the structure that would mean density for residues involved in salt bridges is poorly defined. So a detailed analysis without specific testing would be beyond the scope of this work. Yet, at a minimum, there should be at least some more specific discussion of this point with recent primary references that deal with a structural element that is key to the ATP-driven conformational change in the Rad50 ABC ATPase and likely in this similar ABC ATPase. This, unfortunately, is currently unclear from the manuscript.

2) The authors' response to point 7, asking if there are similarities in the effects of ATP binding to the MutS conformations seen in Hura et al., doesn't seem to make sense. They mention that in both cases it is the nucleotide that drives the change, yet then state that these observations don't correlate. “In this instance we are discussing the nucleotide and not MutL that drives the change whereas Hura et al. discuss the nucleotide rather than DNA driving the change. Hence this doesn't seem to correlate.” However, because both experiments report on nucleotide-driven changes, they should correlate and be describable in terms of noted similarities and/or differences.

3) The MutL N-terminal domain is also an ATPase. Is MutL also bound to nucleotide in their structure? This doesn't seem to be mentioned, and needs to be. It is mentioned that MutL can dimerize upon ATP binding and that the structure of the complex would sterically allow MutL dimerization based on a previously published structure of the N-terminal domain bound to nucleotide. However, this model is buried in Figure 3—figure supplement 1 and the ATP-binding region is not highlighted, which makes it hard to evaluate. Also, this model would suggest that the second MutL would be free to interact with an adjacent MutS dimer on DNA. While this is not seen as a stable complex in the reported gel filtration assays (only MutS-MutL heterotetramers are seen with DNA and AMPPNP), is there previously published evidence for this?

4) In the subsection “Conformation of the MutS sliding clamp” they comment that “This type of ATP-induced tilting … is more often observed upon ATP binding in ABC ATPases, such as ATP transporters, SMCs and RAD50”. And then go on to say in a subsequent sentence that “the extent of the motion and the crossing of the DNA clamp domains of MutS was unexpected”.

This is confusing as firstly it seems to imply that MutS is not an ABC ATPase when, in fact, it is. Also, it could be argued that this observation of subdomain rotations, while perhaps not predicted from earlier work on MutS, fits well with what is known from other ABC ATPases. Therefore, it is important as it bridges a previous gap in our understanding of MutS conformational changes versus other ABC ATPases. It furthermore facilitates future work to uncover a unified mechanism for how ATP drives conformational changes in ABC-ATPases. So a clearer and more unified discussion would be of considerable use to the MutS field but also to the entire ABC-ATPase superfamily. Perhaps they could provide some numbers giving the rotation angles in different systems, to quantify the size of motion they see here with respect to the other systems.

5) They see that the mismatch binding domain becomes disordered in their structure, and propose that both this and the DNA may be present in the ∼35 Angstrom channel they observe. Some quantitation should be included here. Based on the structured volume of this domain and the volume that dsDNA would take up, is the space available in the channel that they see large enough for both the disordered mismatch recognition domain and DNA?

---

## [Author Response]

*1) According to section 7 of the submission guidelines, crystal structures must be deposited and the PDB validation reports must be provided to the referees. The referees should have a chance to examine these reports before the paper is accepted, and the PDB codes should be given in the text*.

The structures have PDB codes: 5AKB, 5AKC, 5AKD. A statement to this extent has been added to the text.

PDB validation reports are included with the revised version.

*2) In the Abstract, the statement about the structure being based on cysteine crosslinking could be misread as indicating that this was a hybrid structure based largely on inferred distances. It would be better to clarify in the Abstract that the desired conformation was trapped by cysteine crosslinking and that crosslinking experiments were also used to validate the molecular interactions inferred from the structure*.

Thanks for the comment; we have adjusted the Abstract to avoid this potential confusion.

*3) Several experiments are presented – FRET, binding, crosslinking – however, in most a control with non-mismatched DNA is missing. There are claims such as* “*dependent on mismatched DNA*” *that could not be made without these controls. Some of the FRET experiments should also be done with non-hydrolyzable ATP in addition to ATP. For the FRET experiments – the data in supplementary material should include donor and acceptor curves, rather than just final FRET differences*.

This is a reasonable point. We have performed many of these controls in the past, but we have now added the references for those more explicitly when we make these statements e.g. to Winkler et al. (for crosslinking) and to Groothuizen et al. (for the SAXS experiments. We have added the requested homoduplex controls for the FRET experiments, Figure 2—figure supplement 2; we've also added the raw data for these experiments in this figure.

The non-hydrolyzable ATP leads to a state that no longer loads onto DNA (as explained by our new state and discussed in the expanded Discussion). Therefore we have not added this control.

*4) The presentation is clear and concise, but it is disappointing that more of the paper was not devoted to describing the crystal structures plus their relationships to transient states, as together they answer many questions. For example, the packing within the crystal structure, as a potential filament on DNA, is of particular interest to many but left as supplement. An expanded discussion should address the transient states, the specificity and effect of MutL binding to MutS, and the new channel resulting from a novel conformation of MutS that reveals a rearrangement of the subunits in the MutS*^*ΔC800*^
*dimer*.

We have extended the Discussion to clarify how we see this structure in the context of the rich MutS literature.

With regard to the protein packing in the crystals, we think this is a crystallographic artefact as the observed packing could not occur in the presence of known MutL dimers. We have changed the text to reflect our ideas better.

*5) Importantly, this work establishes the ATP-bound conformation for MutS dimer with a greater motion than expected from prior MutS structures. In this novel MutS conformation the subunits are tilted across each other by ∼30{degree sign} which explains the stability of the MutS sliding clamp on DNA, compared to the mismatch recognition state. More detail on how this compares to the Rad50 ABC ATPAse ATP-bound conformation would be useful to readers, as the use of ABC ATPases in both mismatch and double-strand break repair has sparked considerable interest. E.g. the ABC ATPase signature helix identified from Rad50 conveys ATP-binding to Rad50 conformations, is this similar or different in the MutS changes in concert with ATP and MutL (Williams GJ et al., ABC ATPase signature helices in Rad50 link nucleotide state to Mre11 interface for DNA repair. Nat Struct Mol Biol. 2011 Apr;18(4):423-31)? Such conformational switching in Rad50 controls pathway progression and the choice between DNA end joining and excision, which may have parallels to these new MutS results or not (Deshpande RA et al., ATP-driven Rad50 conformations regulate DNA tethering, end resection, and ATM checkpoint signaling. EMBO J. 2014 33, 482-500)*.

We have expanded the discussion on the similarities to other ABC ATPases such as RAD50 and the SMCs. The signature helix itself is indeed important (it contains the region identified by Mendillo et al that is buried by the connector domain) and the new position of the connector domain is overlapping with the position of the coiled coil regions in Rad50 in the AMP-PNP bound state.

However, the detailed interactions driving conformational changes seem different, as e.g. the switches identified by Williams et al., and the charge pairs that they observe as important are not conserved.

*6) The authors*' *state that the* “*… sliding clamp state of MutS bound to a MutL domain is highly informative.*” *However their point that* “*it corresponds to a reaction intermediate that occurs during a series of conformational changes triggered by mismatch recognition*” *depends upon putting this work firmly into the context of existing results. Given the novel conformation of MutS and low-resolution coupled to issues from the heavy reliance on Cys mutants, cross-linked and truncated constructs and the absent electron density for the DNA, the authors would do well to better relate this work to other recent biophysical results where feasible, e.g. Gorman J, et al., Single-molecule imaging reveals target-search mechanisms during DNA mismatch repair. Proc Natl Acad Sci U S A. 2012 Nov 6;109(45):E3074-83 and DeRocco VC, Sass LE, Qiu R, Weninger KR, Erie DA. Dynamics of MutS-mismatched DNA complexes are predictive of their repair phenotypes. Biochemistry. 2014 Apr 1;53(12):2043-52*.

We have expanded the discussion of our work with the paper from Gorman as well as more detail how our work resolves conflicts in the smFRET results from Qiu et al. and Jeong et al.

However, the DeRocco paper focuses on differences between different mismatches before ATP addition. We showed previously in a quantitative SPR analysis that after ATP addition all off-rates are similar, despite differences in initial binding (Groothuizen et al. NAR 2014). Therefore we don't think these differences are relevant for the state observed here and therefore we have not discussed this particular paper.

*7) The authors also point out that their presented combination of structural and biophysical methods provides a powerful approach to resolve conformational changes within large and transient protein complexes. Thus, it would be useful to relate this model to recent data on the comprehensive conformational states for MutS and the model for bending then straightening steps in MutS damage recognition. X-ray scattering data was interpreted to suggest that nucleotide binding drives MutSβ conformational changes (Hura GL et al., Comprehensive macromolecular conformations mapped by quantitative SAXS analyses. Nat Methods. 2013 Jun;10(6):453-4.) These new observations would seem to be in general agreement with the SAXS results, e.g.* “*As this is measured in the absence of MutL it suggests that after mismatch binding, ATP is sufficient to induce movement of the connector domain away from the mismatch-recognition position*.”

In this instance we are discussing the nucleotide and not MutL that drives the change whereas Hura et al. discuss the nucleotide rather than DNA driving the change. Hence this doesn't seem to correlate.

*8) For the crosslinked complex, the authors observe a two-step sequence of events, a first increase in FP simultaneous with an increase in FRET due to kinking of the DNA, then a second event increases FP even more but reduces the FRET signal to below starting value in which* “*… the DNA is not kinked any more but kept relatively rigid by the MutL*” *– these data seem to support and extend results from X-ray scattering, which showed DNA bending then straightening upon formation of MutS/MutL complexes (Hura GL et al., DNA conformations in mismatch repair probed in solution by X-ray scattering from gold nanocrystals. Proc Natl Acad Sci U S A. 2013 Oct 22;110(43):17308-13)*.

Indeed, good point; we've added this reference.

*9) They do not observe the DNA in these structures. They claim that either the DNA is disordered (sliding along the channel that is formed) or that it was released. Could this be experimentally examined? Though not complete verification, were the crystals examined to see whether DNA was in the crystals? In one of the crystal forms the channels of at least 3 copies are aligned – if the DNA is in there – it might show fiber scattering along that axis. Do they see this? Would the DNA be able to stack along there (due to length)? Though they may not have a 2:2:1 complex anymore*.

The DNA would fit in our crystals in each case, but unfortunately we do not observe this. We've carefully examined the diffraction patterns, but we really don't observe any fiber scattering in any of the three crystal forms. As stated in the text, it is possible that the sliding clamp diffuses away from the DNA. Alternatively it is still present, but in a disordered state for which we cannot observe any evidence. This type of weak interaction with DNA is in agreement with the single molecule studies that show that the sliding clamp no longer rotates on the DNA once it has made the clamp state (12).

*10) In the Introduction, around the third paragraph, it would be good to help out the uninitiated reader by noting that the newly replicated strand is recognised by the lack of adenine methylation*.

We have adjusted the text to make this point.

*11) In the first paragraph of the Results, it would be good to mention the nature and length of the crosslinker. Until the clarification in subsection* “*MutS sliding clamp recognition by MutL*”*, it could appear that the authors were referring to disulphide bridge formation*.

We have adjusted the text to clarify this point early.

*12) In the second paragraph of the Results, the phrase* “*purification to obtain the protein in two successive cycles*” *should be clarified. Does this means that two purification steps were carried out*?

We have changed the text in order to make clear that we apply two full cycles of cross-linking and purification in order to fully cross-link the material and obtain sufficiently homogeneous material for crystallization.

*13) In the second paragraph of the subsection* “*MutS sliding clamp recognition by MutL*”*, it is not really appropriate to compare the 15.5Å distance between C-alpha atoms of Cys residues with the 18Å distance (really 17.8Å according to the Pierce information) between C atoms in the maleimide ring. In the cross-linked state, there will be three additional bonds on each side (CA-CB-SG-C), giving a significantly larger maximum distance between C-alpha atoms*.

As there may have been a lack of clarity in this section we've rephrased it to state: ‘Since the distance between Cα atoms of crosslinked residue 246C in MutS^ΔC800^ and residue 132 in MutL^LN40^ is ∼15.5 Å it is clear that an 18 Å crosslinker further spaced by the cysteine side-chains cannot enforce the observed position of the connector domain.’

*14) In the legend to*
Figure 2—figure supplement 1*, there should be a capital* “*D*” *in the definition of the difference map coefficients, i.e.* “*mFo-DFc*”*. The authors should always indicate what types of electron density maps are presented (supplementary figures)*.

We've made this change.

[Editors' note: further revisions were requested prior to acceptance, as described below.]

*1) In response to point 5 about how this new work related to the Rad50 ABC ATPase family member (specifically recent NSMB and EMBO papers), they mention that while* “*the signature helix itself is indeed important … and the new position of the connector domain is overlapping with the position of the coiled coil regions in Rad50 in the AMP-PNP bound state*” *the* “*detailed interactions driving conformational changes seem different*” *(i.e. salt-bridge switches are not conserved)*.

*In the text they note similarities to other ABC ATPases, and mention the signature helix, although they do not cite the recent NSMB or EMBO papers that experimentally dissected the role of residues on the signature helix (they rather cite a 2003 review). And in the Discussion they don't discuss whether or not a different set of salt bridge switches may coordinate conformational changes. To be fair this may be complicated by the low-resolution nature of the structure that would mean density for residues involved in salt bridges is poorly defined. So a detailed analysis without specific testing would be beyond the scope of this work. Yet, at a minimum, there should be at least some more specific discussion of this point with recent primary references that deal with a structural element that is key to the ATP-driven conformational change in the Rad50 ABC ATPase and likely in this similar ABC ATPase. This, unfortunately, is currently unclear from the manuscript*.

Previously, we did not address this as MutS is different from RAD50 in this particular region. Hence this discussion will remain speculative, but we agree that it may be informative to be more specific about the differences between these ABC ATPase family members. Therefore we have added the following paragraph:

“In RAD50 this tilting or ‘closing’ motion is transmitted through a ‘signature coupling helix’ via charged interactions with the signature helix (14; 70). This ‘coupling helix’ is found at the beginning of a long stretch (144-767) in Rad50 that includes the coiled coil region and ends in the signature helix. The equivalent region in MutS is only 10 residues long (660-669) and it is disordered in all structures. It is feasible that this 10-residue loop is critical for transmission of the ATP signal, but the details must be different, since the basic residues in the signature helix are not conserved in MutS.”

*2) The authors' response to point 7, asking if there are similarities in the effects of ATP binding to the MutS conformations seen in Hura et al., doesn't seem to make sense. They mention that in both cases it is the nucleotide that drives the change, yet then state that these observations don't correlate.* “*In this instance we are discussing the nucleotide and not MutL that drives the change whereas Hura et al. discuss the nucleotide rather than DNA driving the change. Hence this doesn't seem to correlate.*” *However, because both experiments report on nucleotide-driven changes, they should correlate and be describable in terms of noted similarities and/or differences*.

This is correct, and we've added the Hura et al. reference to further support that ATP binding in the absence of DNA may be sufficient for the formation of a closed clamp state. However, the conformational changes of the connector domain most likely don't change the overall shape of ATP-bound MutS enough to be visible in SAXS analysis. In fact, the movements of the clamps and the mismatch-binding domains may well be uncoupled. This fits with [65] who noted differences in FRET for the mismatch binding domain of ATP-bound MutS in the presence and absence of DNA. We are discussing this now as follows:

“These two movements could potentially be uncoupled. MutS binding to ATP can already cause a closed clamp-like state, (i.e. perform the tilting movement) as shown by SAXS analysis (33), but may possibly not change the conformation of the mismatched binding domain (65), as consequence of the connector domain movement. This would explain how MutS with ATPγS (or with ATP for a mutant that cannot hydrolyse nucleotides (E694A) (35)) could form a closed clamp state that can no longer be loaded onto DNA (13; 21; 35), but nevertheless is not sufficient to bind MutL.”

*3) The MutL N-terminal domain is also an ATPase. Is MutL also bound to nucleotide in their structure? This doesn't seem to be mentioned, and needs to be. It is mentioned that MutL can dimerize upon ATP binding and that the structure of the complex would sterically allow MutL dimerization based on a previously published structure of the N-terminal domain bound to nucleotide. However, this model is buried in*
Figure 3—figure supplement 1
*and the ATP-binding region is not highlighted, which makes it hard to evaluate. Also, this model would suggest that the second MutL would be free to interact with an adjacent MutS dimer on DNA. While this is not seen as a stable complex in the reported gel filtration assays (only MutS-MutL heterotetramers are seen with DNA and AMPPNP), is there previously published evidence for this*?

Indeed, we did not make the nucleotide state of MutL explicit. We have added the following sentence to the text: “Accordingly the MutL monomers have the apo-conformation of residues 80-103 (8) and no density for a nucleotide is visible.”

To our knowledge, there is no published evidence for stable complex formation of MutS/MutL involving multiple dimers of both proteins forming ordered arrangements along DNA.

*4) In the subsection* “*Conformation of the MutS sliding clamp*” *they comment that* “*This type of ATP-induced tilting* … *is more often observed upon ATP binding in ABC ATPases, such as ATP transporters, SMCs and RAD50*”*. And then go on to say in a subsequent sentence that* “*the extent of the motion and the crossing of the DNA clamp domains of MutS was unexpected*”.

*This is confusing as firstly it seems to imply that MutS is not an ABC ATPase when, in fact, it is. Also, it could be argued that this observation of subdomain rotations, while perhaps not predicted from earlier work on MutS, fits well with what is known from other ABC ATPases. Therefore, it is important as it bridges a previous gap in our understanding of MutS conformational changes versus other ABC ATPases. It furthermore facilitates future work to uncover a unified mechanism for how ATP drives conformational changes in ABC-ATPases. So a clearer and more unified discussion would be of considerable use to the MutS field but also to the entire ABC-ATPase superfamily. Perhaps they could provide some numbers giving the rotation angles in different systems, to quantify the size of motion they see here with respect to the other systems*.

We've changed the phrasing to be more clear that we do see the ABC tilt, but that the effects are more dramatic than we had foreseen. We've changed the text to: “Based on comparison to RAD50 (31) we previously predicted a tilting motion (45), and an open-to-closed transition has been supported by deuterium exchange mass spectrometry (56), but the crossing of the clamp domains of MutS and the effect that this has on DNA binding were unexpected.”

*5) They see that the mismatch binding domain becomes disordered in their structure, and propose that both this and the DNA may be present in the ∼35 Angstrom channel they observe. Some quantitation should be included here. Based on the structured volume of this domain and the volume that dsDNA would take up, is the space available in the channel that they see large enough for both the disordered mismatch recognition domain and DNA*?

In our crystal structure both connector domains move away and interact with MutL, and hence the N-terminal mismatch recognition domains both move away from the channel. However the channel is indeed rather wide and we speculate that this complete movement may not be necessary in an asymmetric situation. That said, small adjustments could be necessary to accommodate possible clashes. In the absence of visible DNA there seem to be too many possibilities to model these potential states.